# The polar amplification asymmetry: Role of antarctic surface height

Marc Salzmann[1]

[1]Institute for Meteorology, Universität Leipzig, Vor dem Hospitaltore 1, 04103 Leipzig, Germany

*Correspondence to:* Marc Salzmann (marc.salzmann@uni-leipzig.de)

**Abstract.** Previous studies have attributed an overall weaker (or slower) polar amplification in Antarctica compared to the Arctic to a weaker antarctic surface albedo feedback and also to more efficient ocean heat uptake in the Southern Ocean in combination with antarctic ozone depletion. Here, the role of the antarctic surface height for meridional heat transport and local radiative feedbacks including the surface albedo feedback was investigated based on $CO_2$ doubling experiments in a low-resolution coupled climate model. When Antarctica was assumed to be flat, the north-south asymmetry of the zonal mean top of the atmosphere radiation budget was notably reduced. Doubling $CO_2$ in a flat Antarctica ("flat AA") model setup led to a stronger increase of southern hemispheric poleward atmospheric and oceanic heat transport compared to the base model setup. Based on partial radiative perturbation (PRP) computations it was shown that local radiative feedbacks and an increase of the $CO_2$ forcing in the deeper atmospheric column also contributed to stronger antarctic warming in the flat AA model setup, and the roles of the individual radiative feedbacks are discussed in some detail. A considerable fraction (between 24 and 80% for three consecutive 25-year time slices starting in year 51 and ending in year 126 after $CO_2$ doubling) of the polar amplification asymmetry was explained by the difference in surface height, but the fraction was subject to transient changes, and might to some extent also depend on model uncertainties. In order to arrive at a more reliable estimate of the role of land height for the observed polar amplification asymmetry, additional studies based on ensemble runs from higher resolution models and an improved model setup with a more realistic gradual increase of the $CO_2$ concentration are required.

## 1 Introduction

Surface temperature changes in response to radiative forcings are far from spatially uniform due to local radiative feedbacks and due to atmosphere and ocean dynamics. While anthropogenic greenhouse gases exert the largest radiative forcing at low and mid latitudes, the largest surface temperature increase is observed in the arctic region. Currently, the arctic surface temperature is increasing at more than twice the rate of lower latitudes (Jeffries et al., 2015), and global climate models suggest that by the end of the century the Arctic will have warmed substantially not only in absolute terms, but also compared to the rest of the globe (Collins et al., 2013). Polar amplification of surface temperature change has also been detected based on arctic proxies for past glacial and warm periods (Miller et al., 2010), and it is not limited to the Arctic. In $CO_2$ perturbation experiments for the early Eocene, Lunt et al. (2012) found that several models consistently simulate the greatest warming in the antarctic region due to the lower topography via the lapse rate effect and the change in albedo.

However, in recent time the Arctic has been warming faster than Antarctica, and based on climate model projections it is also expected that arctic amplification will still be notably stronger than antarctic amplification by the end of this century. According to the fifth assessment report (AR5) by the Intergovernmental Panel on Climate Change (IPCC), coupled climate models from the Coupled Model Intercomparison Project phase 5 (CMIP5) yielded an arctic warming of $4.2\pm1.6$ K compared to an antarctic warming of only $1.5\pm0.7$ K for the years 2081-2100 relative to a 1986—2005 reference period in the RCP4.5 radiative forcing stabilization scenario (Collins et al., 2013). The corresponding tropical warming under this scenario was $1.6\pm0.4$ K.

The focus in explaining arctic amplification has long been on the surface albedo feedback and lack of vertical mixing (e.g. Manabe and Wetherald, 1975; Manabe and Stouffer, 1980; Serreze and Francis, 2006). However, several studies have found that arctic amplification was simulated even in models in which the ice albedo feedback is suppressed (Hall, 2004; Alexeev et al., 2005; Graversen and Wang, 2009). While the surface albedo feedback is still considered to be an important contribution to arctic amplification, more recently, other processes have also been identified as being important. In particular, atmospheric feedbacks (e.g. Winton, 2006; Graversen and Wang, 2009; Serreze and Barry, 2011; Pithan and Mauritsen, 2014, Payne et al., 2015, Cronin and Jansen, 2016) and changes in heat transport from lower latitudes to the Arctic (cooling lower latitudes and warming the Arctic) (e.g. Holland and Bitz, 2003; Hall, 2004; Graversen et al., 2008) have both been suggested to be important contributors to arctic amplification.

Since the average antarctic surface height exceeds 2 km mainly due to the presence of thick ice sheets and a compensation of above sea level (positive) and below sea level (negative) bed rock elevation (Fretwell et al., 2013), the ice is thick enough and the surface temperature is low enough so that in many areas melting from above is expected to play a minor role unless temperatures increase dramatically. Even for a fairly substantial $CO_2$ increase, simulated arctic amplification was found to be larger than antarctic amplification due to a weaker antarctic surface albedo feedback already in the early climate model study by Manabe and Stouffer (1980).

Melting from below associated with increased sea surface temperature, on the other hand, is a major concern in part because of the sea level rise associated with pieces breaking off the antarctic ice shield and sliding into the sea (DeConto and Pollard, 2016), although regions which are located far inland and in which the bedrock remains above sea level are not immediately affected. The sea ice in the Southern Ocean, on the other hand, induces a surface albedo feedback in climate change simulations, and in association with the transition from glacials to interglacials it has long been proposed that melting at the edges of ice sheets and glacier flow can successively lower the altitude of the antarctic ice to where it becomes more prone to melting (Hughes, 1975). The corresponding processes are difficult to represent in climate models, and at present, not all climate models include them.

In addition to the surface albedo feedback (Manabe and Stouffer, 1980), the antarctic surface height is expected to influence atmospheric heat transport. Atmospheric feedbacks such as the water vapor feedback are also expected to differ between the Arctic and Antarctic due to the differences in surface elevation. Furthermore, the asymmetric land-sea distribution with more land in the northern hemisphere could potentially play a role in the polar amplification asymmetry.

Another reason for the polar amplification asymmetry that has recently been investigated by Marshall et al. (2014) is an asymmetric response of the ocean heat transport to ozone and greenhouse gas forcing. Masson-Delmotte et al. (2013) have also

indicated that asymmetric warming between the Arctic and Southern Ocean in climate models might be linked to asymmetries in ocean heat uptake and ozone depletion over Antarctica.

Here, the role of the antarctic surface height in the temperature response to an abrupt carbon dioxide doubling is investigated based on idealized sensitivity simulations with a low-resolution three dimensional coupled global climate model in which the surface height was artificially set to 1 m above mean sea level for the entire antarctic continent.

The model, the PRP computations, and the analysis methods are described in the next section. In Sect. 3.1, the zonal mean radiation budget is investigated. The surface temperature response to doubling $CO_2$ is discussed in Sect. 3.2. Changes in meridional heat transport are analyzed in Sect. 3.3 and local radiative feedbacks are analyzed in Sections 3.4 and 3.5. In Sect. 3.6 an attempt is made to compare contributions from local feedbacks, heat transport, and heat storage. The temporal evolution of the coupled runs and potential improvements to the model setup used in the present study are discussed in Sect. 3.7.

## 2   Methods

The Community Earth System Model (CESM) (Hurrell et al., 2013) version 1.0.6 was used in the Community Climate System Model Version 4 (CCSM4) configuration (Gent et al., 2011) to perform a set of idealized 600 year coupled model sensitivity runs. The atmospheric component (using the Community Atmosphere Model (CAM4) physics package) was run at spectral truncation T31 (approximately 3.75 degree horizontal resolution) with 26 vertical layers (due to computational constraints). The ocean component (based on the Parallel Ocean Program version 2 (POP2), Smith et al., 2010) was run in the so-called gx3v7 setup with three degree horizontal resolution and 60 vertical layers. Land ice and snow were simulated by the land surface model (Community Land Model version 4, Lawrence et al., 2011) which includes glacier as a land cover type. Sea ice was simulated using a modified version of the Los Alamos Sea Ice Model version 4 (CICE4, Hunke and Lipscomb, 2008). The coupled model was run in a standard configuration without a dynamic ice sheet model.

In the base control run, the atmospheric carbon dioxide concentration was kept fixed at 284.7 ppmv (for the year 1850). In the base 2xCO2 run, the $CO_2$ concentration was doubled. In addition, two coupled sensitivity runs were performed for which Antarctica was assumed to be flat. These runs are usually referred to as flat AA control run and flat AA 2xCO2 run. Table 1 provides an overview of these coupled runs.

The analysis focuses on the transient response to $CO_2$ doubling during the years 76 to 125 and especially during the years 80 to 109 for which the model was re-run and PRP calculations were performed. While most of the temperature response in the upper ocean to the $CO_2$ perturbation takes place during the initial decades of a 2xCO2 perturbation experiment the deep ocean has not yet reached equilibrium during this period which is similar to the situation in present-day climate and transient future climate change. In the base 2xCO2 run antarctic warming (see Sect. 3.7) became stronger than arctic warming around year 200 (and around year 120 for the flat AA 2xCO2 run) which differs from present-day observations of a stronger polar amplification in the arctic compared to the antarctic region and from the results of shorter CMIP5 simulations which the present study aims to help explain.

In the standard model setup, the radiation subroutine was called every other time step. In order to increase the accuracy of the PRP computations (described below), the coupled model was re-run for the years 76 to 125 calling radiation every (half hour) time step instead of every other time step. This ensured that the net top of the atmosphere radiation in the offline PRP and the coupled runs were consistent, although closing the budgets including atmospheric and oceanic transport and heat

storage remained challenging (this point is further discussed below). The results from these radiation (RAD) re-runs are not identical to the original coupled runs because the more frequent radiation computations trigger a new realization. Many of the main findings of this study are, however, independent of this choice and re-running the entire period would have been computationally expensive. The re-runs differ mainly in their temporal evolution, but the PRP analysis based on these runs yields qualitatively similar results, except that the sum of all feedbacks (which is a small term compared to the individual

feedbacks) is closer to TOA net radiation change in the coupled run and that the contributions of the individual feedbacks and transport terms to the decreasing polar amplification asymmetry differ. The advantage of the RAD re-runs is that the PRP analysis is more exact which makes them more suitable for the budget analysis. The advantage of the base model setup is that it is computationally cheaper to run and thus is better suited for longer integrations.

In addition to these coupled simulations, the radiation code was run in an offline setup to perform a set of standard two-sided

PRP computations as well as several sensitivity computations in order to investigate contributions of local radiative feedbacks. The offline version of the radiative code that is included in the the CESM v1.0.6 distribution was described by Conley et al. (2013). In the original PRP method (Wetherald and Manabe, 1988), radiative feedbacks are estimated by substituting individual fields (such as atmospheric water vapor) from the perturbed (2xCO2) simulation into an offline radiation calculation that takes all other fields from the unperturbed (control) run. In the two-sided PRP method (Colman and McAvaney, 1997), in addition

individual fields from the unperturbed control run are substituted into the offline radiation calculation for the perturbed run. The resulting radiation perturbations are then combined according to:

$$\frac{\delta_{2-1}(R)_x - \delta_{1-2}(R)_x}{2} \tag{1}$$

where $R$ is the net radiation flux at the top of the atmosphere (TOA), i.e. the difference between incoming solar and outgoing solar and terrestrial radiation, $\delta_{2-1}$ is the radiation perturbation from the first (2xCO2 into Control) and $\delta_{1-2}$ from the second

(Control into 2xCO2) substitution for variable $x$ (or variables $x$ for feedbacks in which contributions from several model variables such as cloud liquid and cloud ice concentration are combined into a single feedback).

Here, the surface albedo (ALB), the water vapor (WV), the cloud (CL), and the lapse rate (LR) feedback were computed. Furthermore, the $CO_2$ radiative forcing and the Planck (PL) feedback were computed using the two-sided PRP method. The PL feedback was computed by substituting the surface temperature and adding the surface air temperature difference $\Delta T_s(\lambda, \phi)$

to the atmospheric temperatures $T_s(\lambda, \phi, z)$ at each model layer (where $\Delta T_s$ depends on longitude $\lambda$ and latitude $\phi$). The LR feedback was defined as the difference LR=TA-PL, where TA is the perturbation that is obtained by substituting the surface as well as the atmospheric temperatures.

The radiation code was run offline based on three-hourly instantaneous model output for years 80 to 109 of the coupled simulations. This fairly high temporal sampling frequency in combination with a fairly long period of 30 years led to a rather

smooth and less noisy geographical distribution of cloud feedbacks compared to lower sampling rates and also ensured that the top of the atmosphere energy budget in the PRP runs was consistent with the corresponding coupled runs.

In order to study differences of the LR feedback between the base and the flat AA model setup, a set of additional 30-year PRP sensitivity computations was performed (in addition to the standard PRP computations) based on 73-hourly instantaneous model output. In these sensitivity computations, individual variables from the flat AA setup were substituted in PRP computations for the base setup. For example, a PRP calculation was performed in the base model setup using $T_s$ from the flat AA control and the flat AA 2xCO2 run instead of from the base control and the base 2xCO2 run. An overview of the additional sensitivity computations is given in Table 2. The lower output frequency in the additional sensitivity computations was used since it significantly reduces storage requirements and run time although the PRP computations become less accurate.

Strictly speaking, the cloud radiative response to $CO_2$ doubling and surface warming that is computed using the PRP method is not a "pure feedback" since it is not only modulated by surface temperature changes. Instead, the cloud radiative response also contains contributions from fast cloud responses to the $CO_2$ increase which are independent from surface temperature changes. Since the focus here is not on clouds, the historical term "feedback" is nevertheless used for convenience, even though "response" would be more correct. Furthermore, the feedbacks were not normalized by the surface temperature change which facilitates a straight forward comparison of the actual TOA energy flux differences associated with each of the feedbacks between the base and the flat AA model setup. Normalization by the surface temperature change without taking into account heat storage is only appropriate when two equilibrium states are compared.

In analyzing the model output, meridional atmospheric heat transport (AHT) at latitude $\phi$ was computed by meridionally integrating the difference between the net radiative flux at the TOA ($R$) and the net energy flux at the surface ($F_s$, here also defined as positive downward) from the South Pole to latitude $\phi$:

$$AHT(\phi) = \int_{-\frac{\pi}{2}}^{\phi} \int_{0}^{2\pi} \left(R(\lambda, \phi') - F_s(\lambda, \phi')\right) a^2 \cos(\phi') d\lambda \, d\phi' \tag{2}$$

where the net downward surface flux $F_s = R_s - SHF - LHF - SN$ is the sum of the net downward surface radiation flux ($R_s$), and the sensible and the latent heat flux, and $SN$ is a contribution from snowfall; a is the Earth radius. Ocean heat transport (OHT) is available as a standard diagnostic in the CESM. When computing AHT to the polar regions the integral in Eq. 2 was evaluated individually for each pole, integrating from the pole across all latitudes inside the respective polar circle. The corresponding grid cell edges of the atmosphere grid are located at 66.8°North and South. Oceanic heat transport is diagnosed at 66.6°North and South.

The arctic and antarctic region averages were defined as averages over model grid points that are centered poleward of the respective polar circle. In order to roughly compare local radiative feedbacks and changes of heat transport convergence, the meridional heat transport at the edge of the polar region was simply divided by the area of the polar region.

In the flat Antarctica runs, the land height over Antarctica was set to 1 m for the entire continent.

Heat storage terms (here mainly due to changing ocean heat content and a smaller contribution from sea ice and also a minor contribution from snow) were estimated based on the difference in model state during the first and the last month of

the period under investigation. The regional heat storage term should be balanced by changes of net meridional heat transport convergence and TOA net radiation flux. However, after taking into account heat storage, atmospheric and oceanic transport, and TOA radiation fluxes, the energy budgets were still not balanced especially in the Arctic, and it appears that this difficulty is only in part caused by different atmospheric and oceanic grids. Instead, it could in principle either have been caused by a energy

conservation issue in the model (which can not easily be confirmed based on the present analysis) or else by inaccuracies in the analysis methods used here. However, the (mainly arctic) imbalance was found to be of similar magnitude in the base and the flat AA run, so that the residual terms in the differences between the base and the flat AA model setup were small. Therefore, the energy budget analysis was still considered useful. The imbalance between the net radiation deficit and the sum of the other budget terms (dominated by poleward heat transports) varies between 2 and 5% of the net radiation deficit in the polar regions.

For model evaluation purposes, solar and terrestrial radiation fluxes derived from satellite observations for years 2001 to 2014 were taken from the NASA CERES (Clouds and the Earth's Radiant Energy System,  Wielicki et al., 1996) SYN1deg Edition 3A. The orbit covers latitudes up to about 80°.

## 3    Results

### 3.1    Zonal mean radiation budget

In the polar regions, more energy is emitted to space via long-wave (terrestrial) radiation than supplied by the absorption of solar radiation. This energy deficit is balanced by a radiation energy surplus at the equator and poleward heat transport by the ocean and the atmosphere. The radiation energy deficit is smaller in the antarctic region than in the arctic region. The base model setup (Fig. 1a) and observations (Fig. 1b) show a pronounced arctic-antarctic asymmetry in the magnitude of the outgoing terrestrial radiation flux, which is in line with the higher average surface temperature (and lower average surface

elevation) in the Arctic and also with a larger meridional heat flux from lower latitudes toward the Arctic.

       When Antarctica was assumed to be flat, on the other hand, the asymmetry in the simulated radiation budget decreased markedly (Fig. 1a). In the flat AA run, Antarctica was similar to the Arctic in terms of the zonal mean radiation budget, which implies that the overall poleward heat transport was also more symmetric in the flat AA run. Atmospheric and oceanic meridional heat transport will be further analyzed in Section 3.3.

The slight increase in absorbed shortwave radiation over the antarctic continent in the flat AA run (Fig. 1a) is consistent with increased atmospheric absorption and also includes a minor contribution from an increase in snow albedo as the melting temperature is occasionally approached in the flat AA base run. The latter contribution is non-zero in spite of the fact that the entire antarctic continent remains covered by snow year around (not shown). This is because the snow albedo in the CESM decreases with temperature whenever the melting temperature is approached.

## 3.2 Surface temperature response to doubling $CO_2$

Doubling $CO_2$ in the base model setup led to the well known pattern of a strong polar amplification in the arctic and a less strong polar amplification in antarctic region (Fig. 2, based on the RAD re-runs) that is also found in certain transient climate change experiments. In the flat AA run, on the other hand, the polar amplification was increased in the antarctic region while

it decreased over time in the arctic region (the temporal evolution in the original coupled runs will be discussed in Sect. 3.7).

This decrease of asymmetry is also reflected in Table 3 (based on the original coupled runs), in which the polar warming due to doubling $CO_2$ was analyzed for three consecutive 25-year time slices starting in year 51 and ending in year 126. In the base run, the arctic region warmed on average by $1.40 \pm 0.29$ K (mean of three time slices$\pm$ one standard deviation) more than the antarctic region. In the flat AA run, on the other hand, the difference between the arctic and the antarctic warming was

reduced to $0.59 \pm 0.32$ K. Thus, on average, about $56 \pm 30\%$ of the difference in warming between the arctic and the antarctic region (i.e. $0.81 \pm 0.46$ K of $1.40 \pm 0.29$ K) was explained by the antarctic surface height for the three time slices. However, as evidenced by the large standard deviation, this ratio was not constant in time.

In the first time slice, only 24% of the difference were explained by antarctic surface height, while 64% were explained in the second and 80% in the third time slice. When only antarctic temperature change was taken into account (i.e. the arctic

temperature increase was taken from the base setup while the antarctic temperature increase was taken from the flat AA setup), 73%, 26%, and 42% were explained by antarctic surface height. The differences are explained by a pronounced arctic warming toward the beginning of the flat AA 2xCO2 run compared to the flat AA control run and a weaker subsequent arctic warming (i.e. a cooling relative to the initial warming). This result is based on the original coupled runs and differs from the corresponding result of the RAD re-runs since the temporal evolution of the surface temperature differs although the finding

that polar amplification asymmetry decreases markedly in the flat AA model setup holds in the RAD re-runs as well. Because the original coupled runs reflect the standard configuration of the model, and also because the RAD re-runs are only available for the years 76 to 125, the original coupled runs were chosen for this analysis.

Before discussing the underlying temporal evolution of the zonal mean temperature in the original coupled runs in detail in Sect. 3.7, the reasons for the decreased polar amplification asymmetry in the flat AA model setup (Fig. 2) will be analyzed in

some detail based on the RAD re-runs which generally yield sufficiently similar results to the original runs for this analysis to be useful.

## 3.3 Atmosphere and ocean heat transport

In this section, the zonal mean AHT (Fig. 3) and OHT (Fig. 4) for the years 80-109 are compared between the base and the flat AA model setup and their changes in the corresponding $CO_2$ doubling runs are analyzed based on the RAD re-runs. As

expected, based on Fig. 1a, AHT and OHT were more symmetric in the flat AA model setup than in the base setup (although they can not be expected to become completely symmetric because the overall land mass distribution differs between the northern and the southern hemisphere). In the control runs, the poleward AHT increased in the flat AA model setup compared to the base setup mainly in the southern hemisphere. Poleward OHT across the polar circles increased in the southern hemisphere

and also slightly increased in the northern hemisphere. In the original coupled runs, on the other hand, poleward OHT across the polar circle in the northern hemisphere slightly decreased during this period (Fig. 5). Other findings in this section were not affected by this difference.

Doubling $CO_2$ led to an increased poleward AHT in the base and the flat AA model setup. The increase of the southward AHT across the polar circle in the 2xCO2 runs was larger in the flat AA model setup than in the base model setup, but poleward of 60°S $\Delta$AHT changed sign. At the same time, the increase in southward OHT was maximum around this latitude. This indicates that AHT and OHT are closely linked and that they should not be considered in isolation. Both AHT and OHT contributed to decreasing the asymmetry in the flat AA run compared to the base run, and changes of OHT are not confined to the southern hemisphere especially in the tropics. In summary, for the $CO_2$ doubling experiments, AHT and OHT both contributed to an increased southward heat transport in the flat AA model setup, with OHT changes being more important in the Southern Ocean and AHT "taking over" above the continent.

### 3.4  Local radiative feedbacks

In addition to AHT and OHT, local radiative feedbacks and increased $CO_2$ forcing over Antarctica contributed to the decreased polar amplification asymmetry in the flat AA run. Fig. 6 shows the $CO_2$ radiative forcing as well as various radiative feedbacks for the base and the flat AA model setup for years 80–109 from the PRP computations. The residual (RES) represents the difference between the radiative perturbation from runs in which all variables including the $CO_2$ concentration are replaced simultaneously and the sum of the individual contributions from the feedbacks including the PL feedback and the $CO_2$ forcing. For an identical 3-hour sampling interval, replacing all variables yielded the same radiative perturbation as the corresponding 2xCO2 coupled RAD re-run. The SUM and the RES terms in Fig. 6 are expected to balance the contributions from changes in AHT convergence, OHT convergence, and heat storage. The corresponding individual contributions were diagnosed separately as explained in Section 2 and they were found to be of the same order of magnitude as the SUM term which is much smaller than the individual contributions of the major feedbacks except CL (not shown). Unfortunately, however, as explained in Section 2, the diagnosed contributions do not add up to the SUM+RES term as expected.

The difference between Fig. 6 (a) and (b) is shown in Fig. 7. Maps of the feedbacks in the base model run are provided in Fig. 8. Fig. 9 shows maps of the differences between the feedbacks in the flat AA and the base model setup.

Shorter red than blue bars in Fig. 7 indicate that the difference of the local radiative feedbacks between the arctic and the antarctic region has decreased in the flat AA run. The polar asymmetry for FSU (that is the sum of all feedbacks except PL) has roughly halved. The main contribution was the LR feedback, which will be further analyzed in Sect. 3.5. In addition, the water vapor feedback increased in the antarctic region as expected for a deeper atmospheric column. The surface albedo feedback changed only slightly and the overall contribution of the cloud feedback was small in broad agreement with results from other coupled models (Pithan and Mauritsen, 2014).

The most pronounced surface albedo feedback in the Southern Ocean was found north of the Antarctic Circle (Fig. 8). The more positive feedback in this region in the flat AA run (Fig. 9) contributed to the overall decreased polar amplification asymmetry. The antarctic region as defined here was not directly affected by the sea ice changes north of the Antarctic Circle

(since it was defined as the region south of the Antarctic Circle), but rather indirectly via meridional heat transports. This indicates that the values in the bar charts should not be overinterpreted. Furthermore, the small tropical LR feedback in Fig. 8 is noteworthy which together with the large positive high latitude LR feedback led to a positive global LR feedback during years 80–109 of this particular run. The (model dependent) net cloud feedback over the Pacific warm pool was dominated by a negative short wave feedback rather than the predominantly positive long wave feedback.

In the next section, the large contribution from the LR feedback to the decrease in polar amplification asymmetry is analyzed in some detail. Then the results are combined with the results from the previous sections.

## 3.5  Contribution of surface temperature change to lapse rate feedback

The LR feedback was defined as the difference LR=TA-PL, where PL depends only on the change in surface temperature (which is applied at each height level throughout the atmospheric column) while for TA changes in surface as well as the atmospheric temperatures were taken into account. In the tropics, the atmospheric lapse rate is coupled to the surface temperature via deep convection and via the dependence of the slope of the moist adiabat on surface temperature. In the polar regions, where subsidence takes place and deep convection is absent, atmospheric temperatures are less strongly coupled to surface temperatures, and the LR feedback is less well understood.

Fig. 10 shows that the actual mid- and upper tropospheric temperature change in response to doubling $CO_2$ in the antarctic region was similar in the base and the flat AA model setup, even though assuming a flat Antarctica affected atmospheric dynamics in various ways. This indicates that the large difference in the LR feedback between the flat AA and the base model setup might not only depend on changes in the mid and upper tropospheric temperature profile. Instead, they could have been influenced by different surface temperatures in the base and the flat AA control run. The apparent mismatch between the dots and crosses and the profiles at the lowest atmospheric levels in Fig. 10 (b) and (d) is explained by the condition that all pressure levels where more than 20% of the grid points are above ground are shown in the profiles. Consequently, only a limited number of grid points entered the average temperature for the lowest atmospheric levels while the average surface temperature was computed as an average over all grid points.

In order to find out whether the difference of the LR feedback between the flat AA and the base run can be attributed to differences in atmospheric temperatures or in surface temperatures, a number of PRP sensitivity computations were conducted in which variables from flat AA model runs were substituted into results of runs from the base setup (see Sect. 2 and Table 2 for details). The results from these runs are shown in Fig. 11.

In the first sensitivity experiment (LRPLsens) surface air and atmospheric temperature were taken from the flat AA setup to perform the PRP computations. The LR feedback in this sensitivity computation is similar to the LR feedback in the flat AA standard PRP computation. This indicates that the depth of the atmospheric column was not the main reason for the difference of the LR feedback between the flat AA and the base model setup.

In the second sensitivity computation (LRSens), only the atmospheric temperatures were taken from the flat AA setup, and in the third computation (PLSens) only the surface temperatures were taken from the flat AA setup. When only atmospheric temperatures were replaced, the LR feedback was similarly strong as the one in the base setup. Applying only the surface

temperatures from the flat AA model setup in the base model setup on the other hand explained the stronger antarctic LR feedback in the flat AA model setup, even though the actual atmospheric lapse rate above the lowest atmospheric model layer did not change in the PRP computations.

In other words, the difference of LR=TA-PL between the flat AA and the base run appeared to be dominated by a change of PL and not by a change of TA and it is therefore difficult to interpret the changes in the LR feedback in terms of atmospheric lapse rate changes

Consequently, the LR and the PL feedback will in the next section be considered together. The rationale for this is that in the present study setup the sum LR+PL in the antarctic region is mainly sensitive to changes in surface temperature and not to changes in the atmospheric lapse rate above the surface layer. Note that by definition, LR+PL corresponds to total temperature feedback TA, i.e. the perturbation that is obtained by substituting the surface as well as the atmospheric temperatures. The definition of the TA feedback is identical to the definition of the long-wave feedback in a study by Winton et al. (2006) and to the definition of the temperature feedback in a study by Block and Mauritsen (2013).

The usual decomposition of the total temperature feedback into PL and LR feedbacks is nevertheless useful for understanding polar climate change. As explained above, the PL feedback is defined as the hypothetical feedback that would be expected if the atmosphere would warm at the same rate as the surface. However, the polar atmosphere generally warms less than the surface due to a lack of vertical mixing (e.g. Manabe and Wetherald, 1975). The tropical atmosphere, on the other hand, warms more than the surface. Therefore, in order to radiate away a given amount of energy a larger surface warming is required in the polar regions compared to the tropics (Pithan and Mauritsen, 2014). The lack of atmospheric warming in the polar atmosphere relative to the surface is reflected in the large positive polar lapse rate feedback.

## 3.6   Local radiative forcing, feedbacks, and heat transport

In Sect. 3.3 it was argued that the partitioning between the atmospheric and oceanic heat transport toward Antarctica strongly depends on the exact latitude that is used to define the antarctic region. Furthermore, in Section 3.4 it was argued that the surface albedo feedback outside the antarctic region as defined by the grid points poleward of the Antarctic Circle almost certainly acts to decrease the polar amplification asymmetry in contrast to the surface albedo feedback inside the region. Furthermore, in Section 2 it was explained that closing the energy budgets turned out to be more problematic than anticipated, especially for the Arctic. In spite of these caveats, a comparison between local forcings, feedbacks, changes of heat transport convergence and heat storage terms is attempted here.

Fig. 12 shows essentially the difference between the blue and the red bars in Fig. 7 where PL and LR have been combined into a single temperature feedback based on the arguments in Sect. 3.5 and FSUP is defined as the sum of all feedbacks including the Planck feedback. In addition, the heat transport convergence differences and the heat storage terms are shown. On the whole it appears as if the changes in local feedbacks have increased the asymmetry, although this finding depends on whether or not one chooses to follow the arguments in the previous section and to include the Planck feedback here. Apart from the LR feedback (see Sect. 3.4) which here was combined with the Planck feedback into a single temperature feedback, only the $CO_2$ forcing and the WV feedback acted to decrease the polar amplification asymmetry.

On the other hand, atmospheric changes in heat transport and also in heat content (heat storage terms) appear to have contributed to the decreased asymmetry found in the previous sections. Part of the reason for the change in heat content being fairly important is that the ocean heat content south of the Antarctic Circle decreases in the flat AA base run and increases in the flat AA 2xCO2 run.

This result should, however, not be over-interpreted since for example the albedo feedback most likely would contribute to the overall decrease in asymmetry if contributions from outside the region were also taken into account. Furthermore, in Sect. 3.3, it was argued that ocean heat transport also contributed to an overall decrease in asymmetry since it transports heat to the Southern Ocean where the atmospheric heat transport 'takes over'. Finally, in Sect. 3.3 it was shown that small OHT changes in response to doubling $CO_2$ across the Arctic Circle that contribute to the OHT difference in Fig. 12 depend on the whether the original runs or the RAD re-runs are analyzed.

In a more qualitative sense, Fig. 12 suggests that contributions from changes of heat transport convergence and from heat storage (which would be zero in equilibrium) were of roughly the same magnitude as contributions from local feedbacks, indicating that local feedbacks and changes in heat transport convergence both played a role.

### 3.7 Evolution of surface temperature in the coupled runs

Fig. 13 shows the temporal evolution of the zonal mean surface temperature in the control runs and the 2xCO2 runs for the original coupled runs. As expected, the largest surface temperature response to assuming Antarctica to be flat occurred during the first decades of the flat AA control run (Fig. 13c). These decades were not taken onto account in the preceding analysis. After this, the surface temperatures remained fairly stable in the flat AA control run, although the deep ocean was still adjusting. Taking the difference between the flat AA 2xCO2 run and the flat AA control run (as was done in the preceding sections) is expected to remove most of the latter effect.

Fig. 13b and d show that antarctic surface temperatures increased faster in the flat AA 2xCO2 run than in the base 2xCO2 run as expected based on the previous sections. The arctic temperatures varied strongly due to internal variability, which helps to explain the differences between the original coupled run and the coupled re-runs with half-hourly radiation calls which have been pointed out in the discussion of the ocean heat transport in Sect. 3.3.

The weaker arctic warming in the middle of the 2xCO2 base run (Fig. 13b) is an indication of a slowing of the ocean's meridional overturning circulation (MOC). Such a slowdown has often been found in $CO_2$ perturbation experiments, and it tended to be stronger in low-resolution low-complexity models compared to state-of-the art high-resolution models. Since the CESM was run at a low resolution in this study, this finding should also not be overinterpreted. In the 2xCO2 flat AA run, the MOC started to slow down earlier than in the 2xCO2 base run (Fig. 13d), which might indicate that assuming a flat Antarctica did not only influence the Antarctic region but also the arctic region. For a more reliable estimate of this effect coupled runs at a higher resolution and an ensemble of model runs with slightly perturbed initial conditions would be required.

Fig. 14a shows the evolution of the arctic and antarctic surface temperature for the base and the flat AA model setup and Fig. 14b shows ratios of the arctic to the antarctic amplification which are computed as:

$$f = \frac{A_{Ar}}{A_{AA}} = \frac{\hat{T}_{Ar,2xCO2} - \hat{T}_{Ar,Control}}{\hat{T}_{AA,2xCO2} - \hat{T}_{AA,Control}}$$

where $\hat{T}$ is a regional average 25-year running mean temperature.

It should be noted that antarctic warming relative to the respective control run (Fig. 14a) was stronger in the flat AA than in the base model setup almost throughout the entire 600 year period. However, even though the temperature in the flat AA control run stabilized after a moderate initial warming and even though the temperature evolution from the control run was subtracted in this analysis, it can not be completely ruled out that this moderate initial warming could have also played a role in the later development in the 2xCO2 flat AA run. Therefore, in retrospect, starting the flat AA 2xCO2 from a separate long flat
AA spinup run and prescribing a more realistic gradual increase of the $CO_2$ concentration which would allow to also inspect the first decades of the CO2 perturbation experiments would have been better.

After 250 years, $f$ was lower than unity in the base and the flat AA model setup which indicates that the antarctic temperature increase was stronger than the arctic temperature increase in both model setups. This finding is related to a slowdown of the north Atlantic MOC in both of the 2xCO2 runs which could in part be a transient feature as the MOC recovery times are known
to be extremely long. Again, in order to gain confidence in this result, additional ensemble model runs at higher resolution and ideally also a comparison with results from a multi-model ensemble would be necessary. Since the aim of the various model sensitivity runs has been to investigate the sensitivity of the polar amplification asymmetry to antarctic surface height (and not to provide a future projection of antarctic climate change under global warming), the runs were performed without an ice sheet model. In order to arrive at a more credible projection of antarctic climate change, state-of-the art high resolution models that
include state-of-the art ice sheet dynamics models should be used.

## 4   Conclusions

Idealized $CO_2$ doubling experiments in a low-resolution coupled climate model were performed in order to investigate the effects of antarctic surface height on the polar amplification asymmetry. It was found that antarctic surface height indeed played an important role in the slower antarctic warming as expected based on the early climate model study by Manabe and
Stouffer (1980). Furthermore, it was found that assuming Antarctica to be flat strongly reduced the hemispheric asymmetry of the zonal mean top of the atmosphere radiation budget, which already by itself indicates that meridional heat transport was also more symmetric in the flat AA model setup.

The polar amplification in the 2xCO2 runs also became notably more symmetric when Antarctica was assumed to be flat. In addition to meridional heat transport, also the stronger $CO_2$ radiative forcing and the stronger water vapor feedback over
Antarctica in the flat AA runs contributed to the decrease in asymmetry. All of these changes were expected in the deeper atmospheric column that resulted from assuming Antarctica to be flat. For ocean heat transport and the albedo feedback it was argued that also changes north of the Antarctic Circle had to be taken into account when assessing their contribution to

the decreased polar amplification asymmetry in the flat AA runs. Both decreased rather than increased the polar amplification asymmetry in the flat AA 2xCO2 run.

Among the local radiative feedbacks, the biggest contributor to the decreased polar amplification asymmetry was the lapse rate feedback, in agreement with Lunt et al. (2012). However, based on results from additional experiments, it was argued that in this particular model setup, the change of the lapse rate feedback was mainly linked to the change in surface temperature and much less dependent on changes in the atmosphere above the surface. Therefore, it was argued that one might combine the lapse rate and the Planck feedback into a single temperature feedback. The disadvantage of this approach is that it blurs the distinction between the Planck and the lapse rate feedback. Although the rationale for decomposing the total temperature feedback into the Planck and the lapse rate feedback is less clear in the polar regions than in the tropics, such a decomposition is nevertheless useful for understanding polar climate change (see e.g. Pithan and Mauritsen, 2014; Payne et al., 2015). A more detailed discussion of this issue was given at the end of Sect. 3.5.

Other important factors such as stratospheric ozone depletion (Masson-Delmotte et al., 2013) that were found to contribute to the polar amplification asymmetry in previous studies were not investigated in this study.

Given the important role of increased atmospheric heat transport in the flat AA runs in the present study, one could argue that a decrease in land height due to antarctic melting would be favorable for increased atmospheric heat transport from mid latitudes. Consequently, once the antarctic surface height is lowered due to melting, it would be more difficult to restore the antarctic ice sheet due to increased atmospheric heat transport.

Finally, it was found that assuming a flat Antarctica may not only influence the antarctic but also the arctic region. However, in order to arrive at more reliable results regarding this point, one would have to perform an ensemble of higher resolution model runs because it has long been understood that the MOC can be overly sensitive to perturbations in low-resolution coupled models. Potential future studies based on coupled climate model runs that aim to study the influence of surface elevation on the polar amplification asymmetry which is found in present-day observations and also in future climate projections (which often span only one or one and a half centuries) would benefit from a separate long flat AA spinup run and from prescribing a more realistic gradual increase of the $CO_2$ concentration.

# 5 Code availability

The CESM model code is available from http://www2.cesm.ucar.edu.

# 6 Data availability

The CERES satellite data product can be obtained from http://ceres.larc.nasa.gov/order_data.php. Model output used in this study can be provided by the author upon request.

*Acknowledgements.* I would like to thank the reviewer Timothy W. Cronin and an Anonymous Reviewer for many insightful and constructive comments and I would like to thank the developers of the CESM and the NASA CERES teams at the NASA Langley Science Directorate. The CESM is maintained by the Climate and Global Dynamics Laboratory (CGD) at the National Center for Atmospheric Research (NCAR) and is sponsored by the National Science Foundation (NSF) and the U.S. Department of Energy (DOE). I would also like to thank Johannes

5 Mülmenstädt, Johannes Quaas, and several other colleagues for useful discussions. I am especially grateful for support by the German Research Foundation (DFG) in TRR 172 "ArctiC Amplification: Climate Relevant Atmospheric and SurfaCe Processes, and Feedback Mechanisms, (AC)[3]", sub-project E01 "Assessment of Arctic feedback processes in climate models" (INST 268/331-1).

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

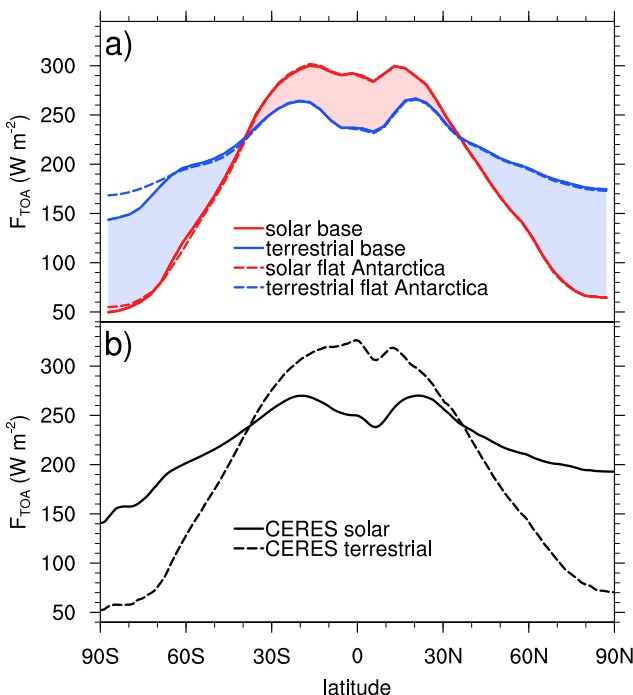

**Figure 1.** (a) Zonal mean radiation budget at the top of the atmosphere (TOA) for years 80 to 109 of the base and flat AA control run: net downward solar (= absorbed shortwave, red lines) and upward terrestrial (= outgoing longwave, blue lines) radiation flux. The shaded areas are based on the base control run. Red shading indicates a net radiation surplus (i.e. on the net, more solar radiation is being absorbed than terrestrial radiation emitted) and blue shading a net radiation deficit. Since the focus is on the polar regions, the x-axis has not been scaled by the cosine of the latitude. (b) Zonal mean radiation budget at the top of the atmosphere (TOA) for years 2001 to 2014 based on the CERES SYN1deg Edition 3A satellite data product.

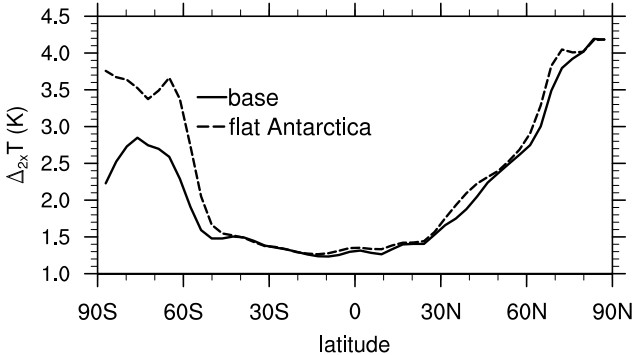

**Figure 2.** Surface air temperature increase for a $CO_2$ doubling in the base and the flat AA model setup for years 80-109 (from the RAD re-runs).

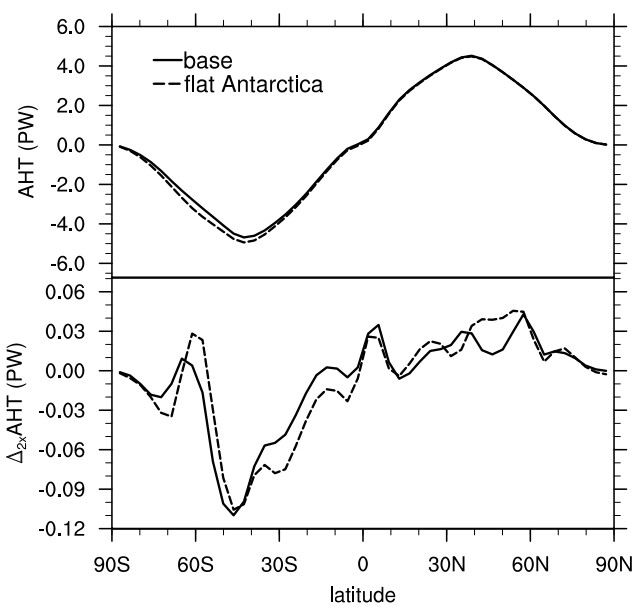

**Figure 3.** (a) Northward atmospheric heat transport (AHT) in petawatt in the base and the flat AA model setup for years 80-109. (b) Differences (2xCO2 minus Control) for doubling $CO_2$ (from the RAD re-runs).

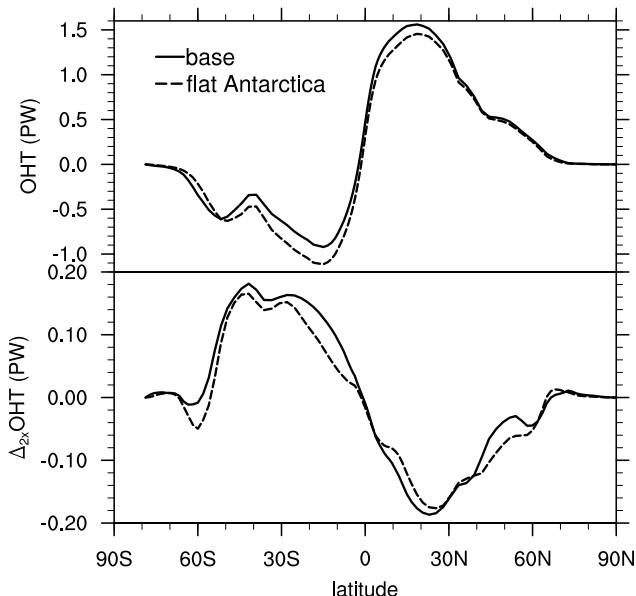

**Figure 4.** Same as Fig. 3 but for oceanic heat transport (OHT).

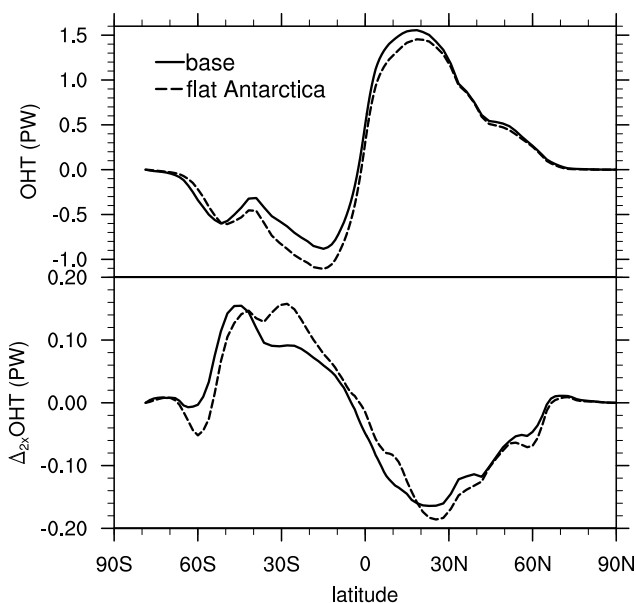

**Figure 5.** Same as Fig. 4 but from the original runs instead of the RAD re-runs.

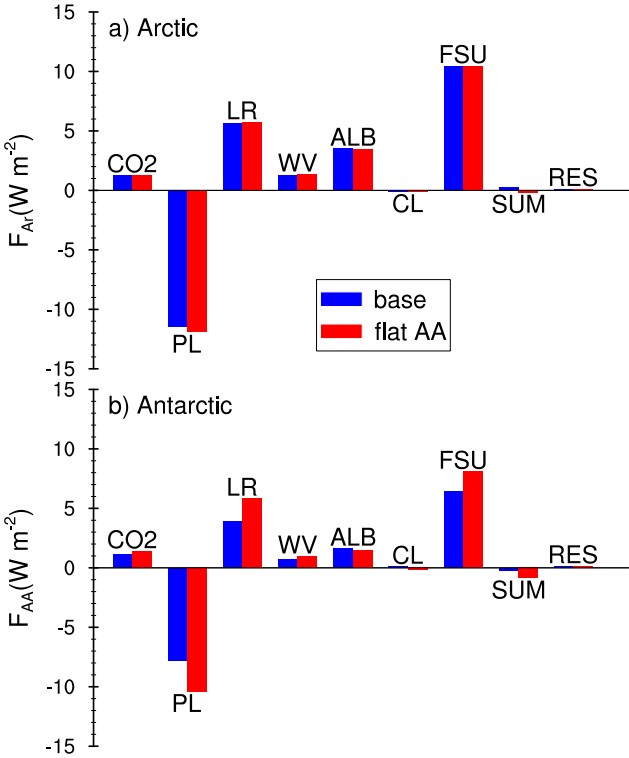

**Figure 6.** $CO_2$ radiative forcing (CO2) as well as Planck (PL), lapse rate (LR), water vapor (WV), albedo (ALB), and cloud (CL) radiative feedback based on two-sided PRP calculations for years 80–109. FSU=LR+WV+ALB+CL is the sum of the feedbacks except PL. The residual (RES) is the difference between the radiative perturbation from replacing all variables simultaneously and the sum (SUM=CO2+LR+WV+ALB+CL+PL) of the individual contributions including CO2 and PL.

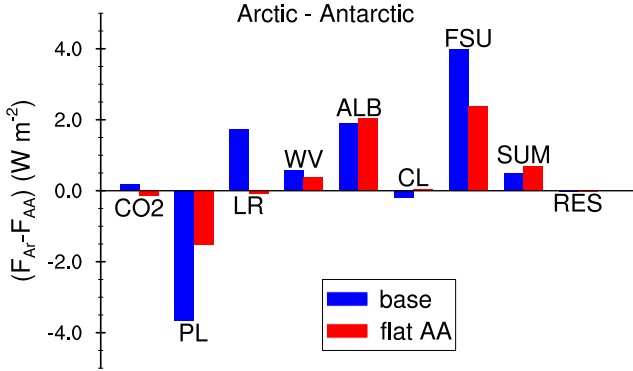

**Figure 7.** Differences between Fig. 6 (a) and (b).

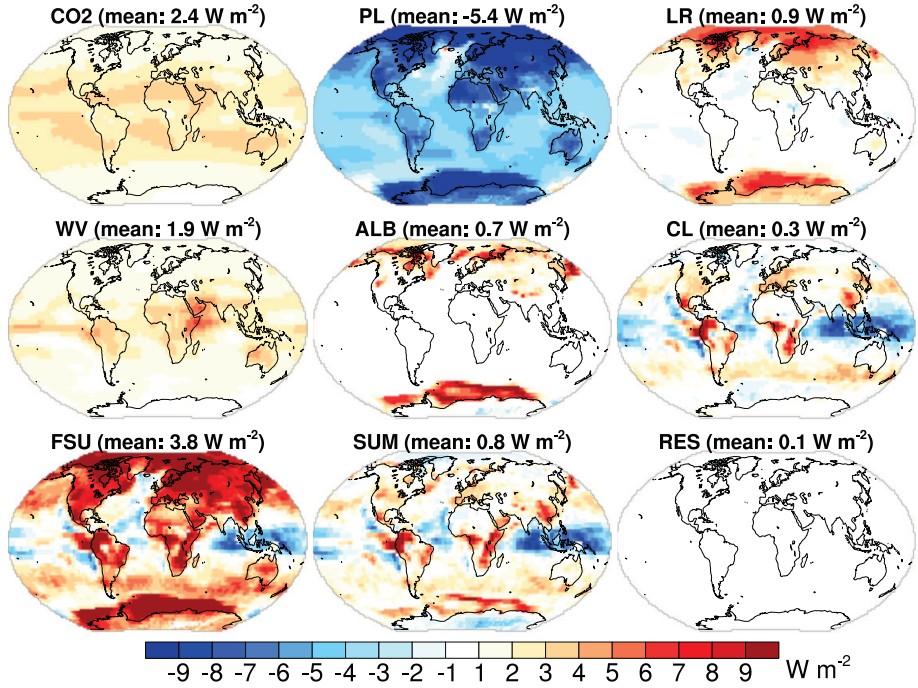

**Figure 8.** Radiative forcing and feedback maps for the base model setup for the years 80–109. Labels as in Fig. 6.

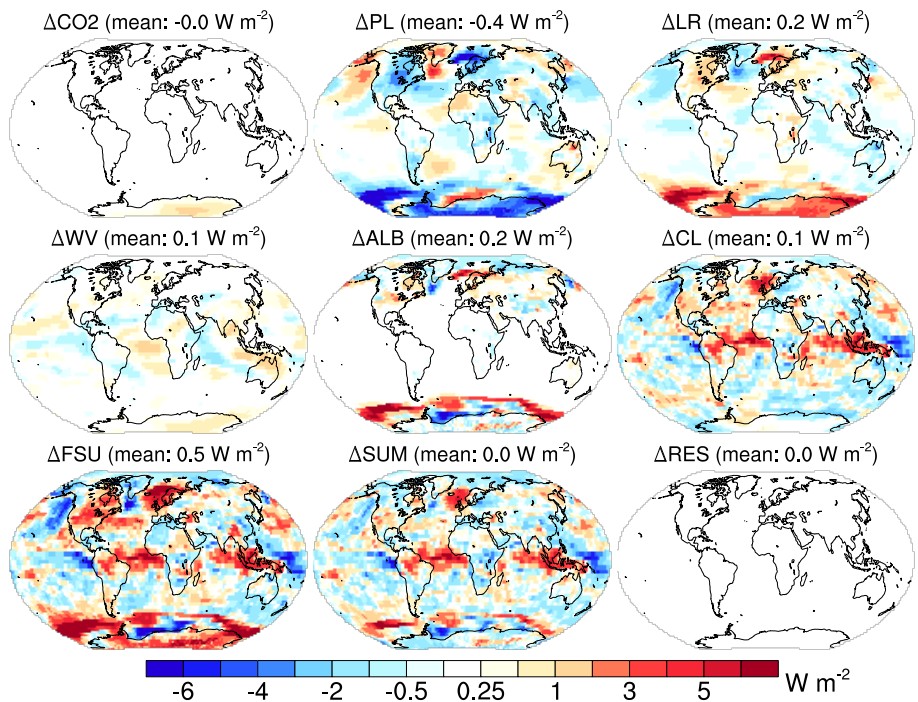

**Figure 9.** Differences between the flat AA and base the setup (comparable to Fig. 8).

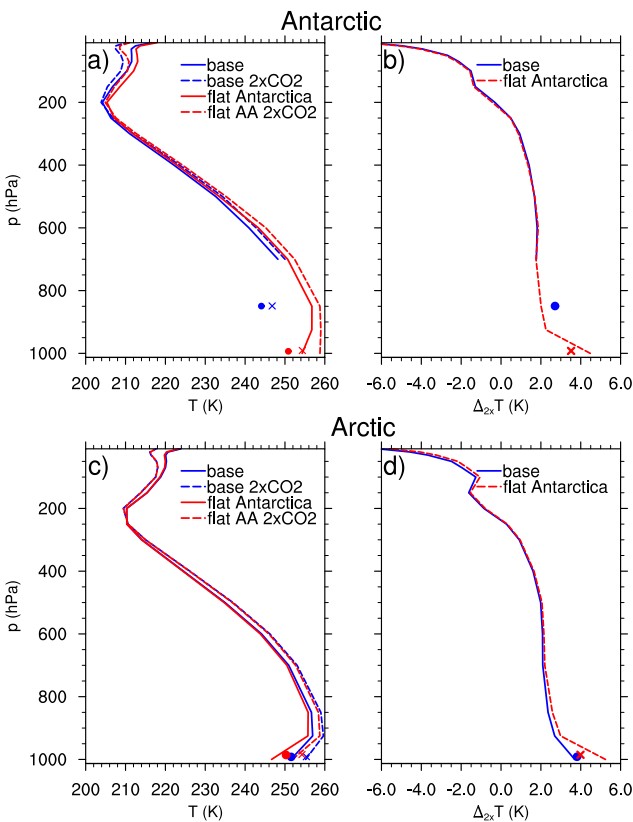

**Figure 10.** Average air temperature profiles for the antarctic and the arctic region from the coupled runs and differences (2xCO2 minus Control) for the base and the flat AA model setup for years 80–109. Dots and crosses denote average surface air temperature at the region average surface pressure. Dots correspond to solid and crosses to dashed lines. Only pressure levels where more than 20% of the grid points are above the ground are shown in the profiles.

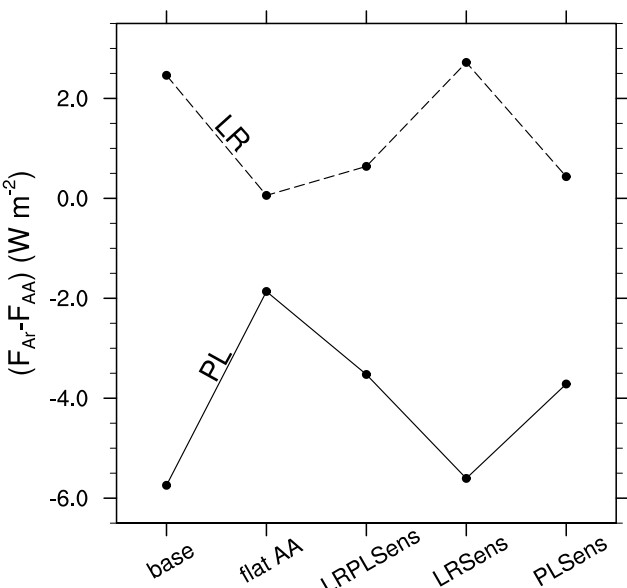

**Figure 11.** Arctic minus Antarctic difference of the lapse rate (LR) and Planck (PL) feedback in the base and the flat AA model setup and for the sensitivity calculations described in Table 2. All computations are based on PRP calculations using 73-hourly instantaneous model output and the original base and flat AA runs are analyzed. .

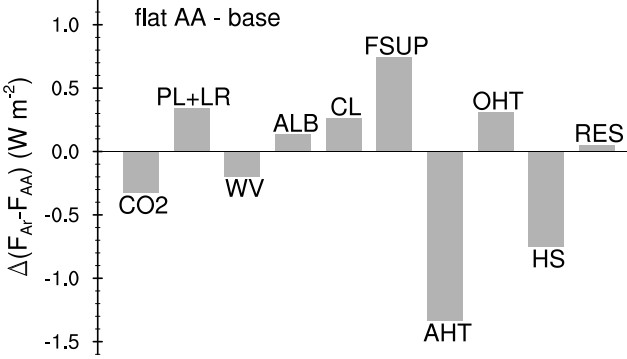

**Figure 12.** Difference flat AA minus base setup of $(F_{AR} - F_{AA})$, where $(F_{AR} - F_{AA})$ is the difference between the arctic and the antarctic region shown in Fig. 7. For the forcing and the feedbacks, this corresponds to the difference between the red and the blue bars in Fig. 7. FSUP=PL+LR+WV+ALB+CL is the sum of all feedbacks including the Planck feedback. It is compared to contributions from atmospheric and oceanic heat transport (AHT, OHT) and heat storage (HS). The residual (RES) is defined as the difference between HS and the sum of forcing, feedbacks and heat transport convergence (CO2+FSUP+AHT+OHT).

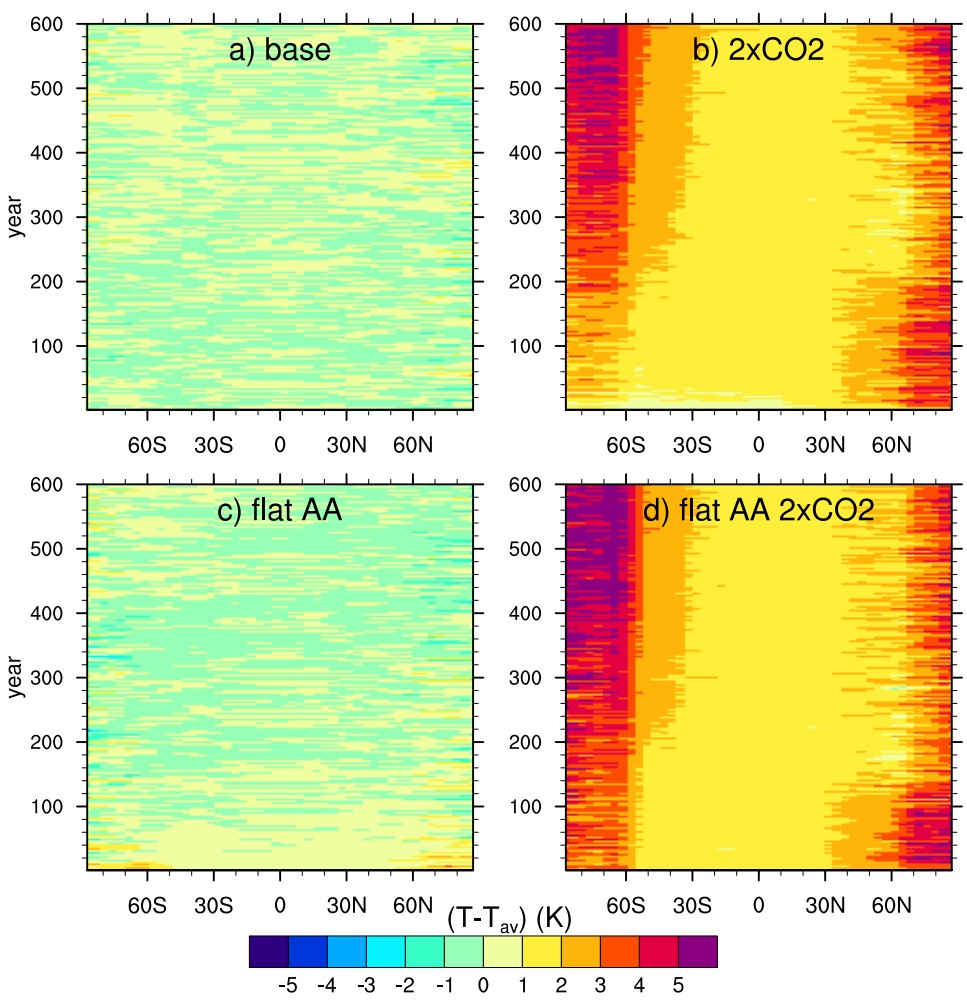

**Figure 13.** Difference between the zonal mean surface air temperature and the time averaged zonal mean surface air temperature from the corresponding control run ($T_{av}$) in the control and the 2xCO2 run for the base and the flat AA model setup.

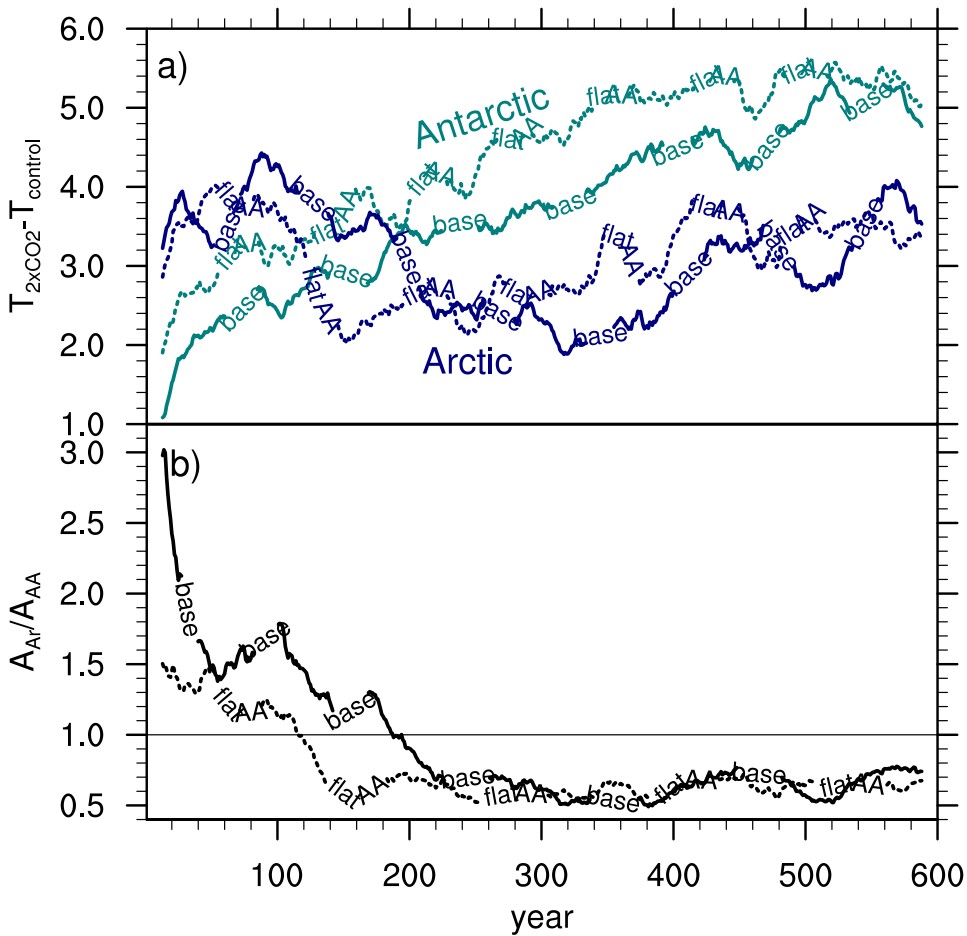

**Figure 14.** (a) 25-year running mean of the temperature difference between the 2xCO2 and the control runs for the arctic and the antarctic region in the base and the flat AA model setup and (b) ratio of arctic to antarctic amplification based on 25 year running mean time series.

**Table 1.** Coupled Runs

| run | description |
|---|---|
| base Control | Control run for constant year 1850 conditions |
| base 2xCO2 | Same as base Control, but doubled $CO_2$ concentration |
| flat Antarctica Control | Control run for flat Antarctica (flat AA) |
| flat Antarctica 2xCO2 | flat AA $CO_2$ doubling run |

**Table 2.** Additional PRP Sensitivity Computations

| label | variable(s) from flat AA model setup in base model setup |
|---|---|
| LRPLSens | surface air ($T_s$) and atmospheric ($T_a$) temperature |
| LRsens | atmospheric temperature $T_a$ |
| PLSens | $T_s$ and control $T_a$ with added $\Delta T_s$ as in PL |

**Table 3.** Mean arctic and antarctic warming (2xCO2 minus Control) $\pm$ one standard deviation for three consecutive 25-year time slices starting at year 51 for the base and the flat AA model setup.

| | Arctic Warming (K) | Antarctic Warming (K) | Difference |
|---|---|---|---|
| base | $3.95 \pm 0.48$ | $2.55 \pm 0.22$ | $1.40 \pm 0.29$ |
| flat AA | $3.76 \pm 0.31$ | $3.17 \pm 0.01$ | $0.59 \pm 0.32$ |