# Peer review of "The polar amplification asymmetry: Role of antarctic surface height"

_Earth System Dynamics, 2016_

## Referee Comment (RC1) · Anonymous Referee #1 · 23 Jan 2017

The paper examines the role of Antarctic surface height for for the asymmetry of polar amplification using dedicated model runs with a standard continental setup and with a flat Antarctic continent and ice sheet. The question is very relevant and the results are interesting, so I recommend the paper for publication after the following issues have been addressed:

Major issues: 1) differences between RAD and ctrl runs and temporal evolution Going back and forth between results from both types of runs at times becomes confusing for the reader. I suggest a more explicit discussion of the use of both runs and why one of them would be more reliable for a certain analysis than the other in the methods section. Are results that are not robust between these two sets of runs robust enough to be mentioned at all? To the extent that the RAD re-run mostly triggers a new realisation, I would doubt that. Especially for the temporal evolution section, the robustness of

the results has to be demonstrated. 2) Discussion of the role of sfc vs. atmospheric temperatures The author convincingly shows that the change in sfc temperature and its relationship to atmospheric warming is causing changes in the lapse rate feedback, rather than the warming profile within the atmosphere. However, it is not clear to me that the temperature feedback should therefore be regarded as a single mechanism. This point might benefit from either rethinking or more detailed discussion.

Minor issues:

l. 5 if->when (If + past tense triggers would in main clause, and indicates a hypothetical case) This issue reappears in the manuscript. l.6 (and elsewhere): please reserve "significant" for its statistical meaning, and specify the statistical test. p.2 l1 ff: What is the baseline for warming? p.2 l.23 Arctic or Antarctic? p. 2 l.32, general issue: I would suggest to consistently speak of either polar amplification symmetry or asymmetry to avoid confusion. p. 6 l.27 explained by what? p. 11 l. 15 Arctic or Antarctic?

Figure 1 and 2: I suggest combining these into one Figure, or at least using the same axes to facilitate comparisons.

---

## Referee Comment (RC2) · T. Cronin (Referee) · 3 Feb 2017

Earth System Dynamics manuscript esd-2016-74: Salzmann, "The polar amplification asymmetry: Role of antarctic surface height"

reviewed by Timothy W. Cronin

Summary:

This paper uses a set of low-resolution coupled climate model simulations to address the mechanisms that underlie hemispheric asymmetry in polar amplification – or, why the Arctic warms more rapidly and more than the Antarctic. The major finding is that flattening Antarctica reduces the polar amplification asymmetry, largely through the role of increasing poleward atmospheric heat transport in the southern hemisphere

that is linked to ice-albedo feedback just outside the south polar cap. Local feedback differences between north and south poles have mixed roles in altering the polar amplification asymmetry.

Overall, I found the question addressed to be important, the writing to be concise, and the conclusions to be mostly well-supported and novel. I do think, however, that there are a number of issues that require further clarification and discussion. I recommend publication if these can be addressed.

Major Comments:

1) One major issue is that the ocean circulation may change considerably between flat-AA and base setups, and it may not be in pre-industrial equilibrium in either case. A hint of this is seen in Figure 5, and a trace of its implications may also emerge in figure 10. How long was the model run for to get a stable "preindustrial climate" in both the flat-AA and base setups? If the ocean is not equilibrated – and I doubt it would be given that most runs appear to occur for ∼100-200 years and this is an order of magnitude less time than required for equilibration – how do we definitively interpret differences between flat-AA and base setups as being results of changes in terrain and not ocean transients? Some issues with this are hinted at in section 3.6, where southern hemispheric heat content changes are noted as large, and also in section 3.7, where the MOC starts to collapse in the 2xCO2 run and leads to changes in Arctic temperature that may be purely due to internal variability rather than forcing or changes in topography.

2) I find the multiple layers of differences to make some figures, discussion, and notation very confusing. There are essentially three sets of differences used in this paper: a climate change difference (2xCO2 – 1xCO2), a hemispheric difference (Arctic – Antarctic), and a model setup difference (Flat-AA – base). Each of these is at some point annotated as Δ: in Figures 3-5 and 10, a single Δ means a climate change signal, in Figures 7 and 11, a single Δ refers to a Arctic-Antarctic difference, and in figure

9 a single $\Delta$ refers to Flat-AA minus base difference. In the worst case, all three are used together in figure 12 as $\Delta(\Delta F)$, or the difference (flat-AA – base) between Arctic and Antarctic feedbacks (with feedbacks F being in turn a difference between 1x and 2x CO2 radiative fluxes). I would suggest using subscripts for all but one of these differences, e.g., T2x-T1x for climate change in response to doubling CO2, and FA-FAA for hemispheric difference, and reserving $\Delta$ for Flat-AA minus base, as it is least-easily subscripted (though F_ - Fˆ could serve here too).

3) The definition of "feedbacks" as changes in TOA radiative fluxes, un-normalized by surface temperature changes, complicates interpretation of findings in this paper for me. I am not sure what the basis is for this choice is, and I think it could be better explained if there is some strong reason for using these changes rather than the standard of feedbacks being normalized by temperature change – e.g., $\lambda$ = (R2x – R1x)/(T2x – T1x) (units of W m-2 K-1). In Figures 6 and 7, especially, I find that use of changes in TOA fluxes rather than standard feedbacks convolutes changes in polar warming with changes in the true per-unit warming feedbacks $\lambda$. For example, do the Planck and lapse-rate "feedbacks" in Figure 6 change just because there is more warming in the Antarctic in the flat-AA case, or because each $\lambda$ has also changed? Use of the conventional feedback metric normalized by surface temperature change would also allow for more straightforward comparison to previous work.

4) The reasons for the RAD re-runs are not made completely clear, and it is not clear where they are used and where they are not used. This is a particular issue around line 30 on page 6, where you say "the original coupled runs were chosen for this analysis" but then in the next paragraph, you say that the simulations behind figure 3 will be analyzed in more detail – but the caption of figure 3 says that they are RAD re-runs.

5) [Self-promotion disclosure!] Some work that I have been involved with recently has, I think, helped provide a better understanding of the high-latitude lapse-rate feedback. You might want to look at Payne, Jansen, and Cronin (2015), GRL, "Conceptual model analysis of the influence of temperature feedbacks on polar amplification" and Cronin

[Figure]

& Jansen (2016), GRL, "Analytic radiative-advective equilibrium as a model for high-latitude climate". Both papers attempt to develop a better prognostic understanding of the high-latitude lapse-rate feedback, which, as you note in section 3.5, is mostly connected to near-surface temperature changes being larger than mid- and upper-tropospheric temperature changes. Figures 3 and 4 of Payne et al (2015) are similar to your Figure 10, except that we focused on low-latitude/high-latitude differences and not polar asymmetries, and we were using simpler column models to get a sense of temperature structure changes. I have also done some preliminary work on polar amplification asymmetry due to surface height with a radiative-advective equilibrium model (nowhere near publication, so certainly nothing to cite!). If you plan to work on the subject more, it might be good to talk outside the review process, and potentially collaborate.

Other specific comments:

A) On page 5, lines 8 and 10 – what are the "respective polar circles", exactly? It's not obvious that you can choose 66.55 N and S when the model's resolution is only T31 – so exact limits of integration and averaging should be noted more clearly.

B) I suggest combining Figures 1 and 2; it is difficult to compare the simulations to observations at present and an additional set of black lines in 1) should be completely readable.

C) Regarding the three time slices and standard deviations in Table 3 and p6, lines 17-27. I don't understand why these three time slices were chosen, or what the % of warming difference explained by topography means, exactly. And how are the +/- standard deviations determined – based on three points (the means of the three time slices), or based on each year treated as an independent point?

D) In Figures 6 and 7: whether or not you stick to calling changes in radiative fluxes feedbacks or you decide to normalize them by temperature changes and make them true feedbacks, I think it would help to include changes in AHT and OHT convergences

here, as well as storage if it is significant. Or, you could mention that the 'SUM' term in each plot adds up to the sum of changes in AHT convergence, OHT convergence, and storage. It surprises me that the changes in heat flux convergence would be so small relative to local feedbacks. I do appreciate your note on p8 L5 that the values should not be over-interpreted as heat transports vary rapidly near the margins of the Arctic and Antarctic caps.

E) Figure 10: The mismatch between the profiles and dots/crosses is confusing here. Why is there so much less change in the crosses in b) and d) than in the profiles at lowest-levels? Why is the red dot in c) warmer than the profile at lowest levels? And why are the base and flat-AA profiles so different in c)?

F) Page 8, paragraph around line 25: I suggest eliminating Table 2 – I found it more confusing than helpful – and relocating the paragraph from the methods section describing these simulations to this point in section 3.5 (paragraph from p4 L20-25). The information is not necessary when presented at the time in section 2, and will be more helpful in this section if the reader does not need to flip back to the earlier description.

G) Page 8 Line 34-Page 9 line 2, re: "weaker LR feedback in the flat AA model setup" – I thought Figure 6 showed the LR feedback was stronger in the flat-AA model setup? Do you mean that the LR feedback asymmetry is smaller in the flat-AA model setup (shown in Figure 7)?

H) Page 9, lines 3-9. I'm a little puzzled by these assertions and the choice to lump the Planck and LR feedbacks together. I agree that your sensitivity tests have shown that the lapse rate feedback is dominated by changes in surface temperature rather than atmospheric temperature, or that the "lapse-rate feedback" is a feedback related to how surface temperatures and near-surface temperatures change relative to the mid- and upper-troposphere (essentially, the lapse rate feedback relates to changes in the strength of surface inversions). Thus, it still seems meaningful to me to separate the lapse rate and Planck feedbacks, since the Planck feedback could be calculated

given the model's control state alone, whereas changes in inversion strength are a complicated result of changes in many linked model processes. [Also, note: I do not see any references to Figure 11 in this section (or elsewhere). Perhaps the reference disappeared at the end of p9 L5? Figure 11 did not seem like the most compelling support for your argument; I think looking at each feedback by itself rather than their hemispheric asymmetry would make a stronger point]

l) Relatedly, on p 11, lines 1-4: I don't really follow this argument, as noted above. I also disagree with this assertion that there is no reason to separate the high-latitude Planck and lapse rate feedbacks. The reason for low-latitude separation is that we can calculate them separately, basically from first principles, if stratification follows a moist adiabat and we know the control-state climate – and further, that there is cancellation between water vapor and LR feedbacks in the tropics. That we don't understand the high-latitude lapse rate feedback fully seems to be a very good reason not to lump it in with a feedback that we do understand better (the Planck feedback).

Line-by line comments:

p1 L 6; p6 L13; p7 L10 – should be "led to" if in the past tense

p1, and then recurring: I am unsure of the correct style guidelines for capitalization of "Arctic", "Antarctic", and phrases containing these two terms. I have (perhaps) erred on the side of always capitalizing. Super minor issue, will presumably be fixed at the proof level, but I am interested in the "right" answer!

p 1 L19 – may want to put numbers on "substantially", e.g., "2-3 times as much as the global average"

p2 L4 – suggest rewording to "The focus in explaining arctic . . . has long been on the . . ." (rather than starting sentence with "For long")

p2 L9; p8 L17 – You could mention work from point 5) above at either of these locations.

p2 L26 – should be "water vapor feedback" (not just "vapor feedback")

p2 L27 – should be "Antarctic" and "land-sea distribution"

p2 L28 – suggest changing "play a role for…" to "play a role in …"

p3 L19-21 – suggest changing "every second time step" to "every other time step" (as seconds are units of time

p4 L16 – suggest hyphenating "three-hourly"

p4 L21-22 – what does "73 hourly instantaneous model output" mean? (This looks like a typo to me)

p5 eqn 2 – use lower-case d's in $d\lambda$ $d\varphi$' ? Also suggest using 'a' for Earth's radius, as R has been used to indicate radiative fluxes

p5 L 16-18 – I believe I had heard, though I am not sure where from, that there is a small added ocean heat source in some variants of CCSM, to keep sea ice from growing too thick. This might contribute to your Arctic energy imbalance, though I am not sure about its magnitude – you might want to reassure the reader here by saying something about the size of the imbalance.

p5 L31; p6 L4 – suggest deleting "rather" in these sentences, as similar is already qualitative

p7 L7-9 – re: "Poleward OHT increased in the southern hemisphere and also slightly increased in the northern hemisphere" and subsequent sentence – Figure 5 seems to me to show a decrease in poleward OHT in the NH… are you showing the "original coupled run" results in Figure 5? (this ties to comment 4 above)

p7 L13 – No comma after "Both"

p7 L29-30 – The interpretation of reduced polar feedback asymmetry is complicated by the feedbacks being expressed as radiative flux changes rather than true feedbacks per unit temperature change (see point 3 above)

[Figure]

p8 L3 – suggest changing to "was not directly affected by. . ."

p9 L 30 – Paragraph around here makes an excellent point about the semi-arbitrary extent of polar caps.

p10 L8, 12 – suggest hyphenating "half-hourly" "low-complexity", and "high-resolution"

————————————————

---

## Author Comment (AC1) · 14 Mar 2017

**Response to Anonymous Reviewer #1**

I very much thank the reviewer for insightful and constructive comments. The reviewer's comments are repeated in blue and the replies are typeset in black. Appended to this response is a draft version of a revised manuscript including track changes (the editor will decide later whether a revised manuscript should be submitted). For brevity, this will be referred to simply as "revised manuscript".

Major issues:

1) differences between RAD and ctrl runs and temporal evolution
Going back and forth between results from both types of runs at times becomes confusing for the reader. I suggest a more explicit discussion of the use of both runs and why one of them would be more reliable for a certain analysis than the other in the methods section. Are results that are not robust between these two sets of runs robust enough to be mentioned at all? To the extent that the RAD re-run mostly triggers a new realisation, I would doubt that. Especially for the temporal evolution section, the robustness of the results has to be demonstrated.

Unfortunately, the RAD re-runs are needed for the budget analysis. On the whole, the original runs and the RAD-reruns are similar, and in the revised manuscript, I have tried to make this clearer and also to more clearly point out where differences exist and to better explain them.

The following sentences have been added to page 4, lines 14ff of the revised manuscript in order to more explicitly explain why two sets of runs are used:

   "The advantage of the RAD re-runs is that the PRP analysis is more exact which makes them more suitable for a budget analysis. The advantage of the base model setup is that it is computationally cheaper to run and thus is better suited for longer integrations."

On page 7 in lines 31f of the revised manuscript it has been stressed that the RAD re-runs are sufficiently similar to the original runs for the budget analysis based on the RAD re-runs to be useful. This point is discussed in more detail in the reply to the major point 4 by the other reviewer Prof. Timothy Cronin.

On page 7 in lines 26f of the revised manuscript the sentence:

*"This result is based on the original coupled runs and differs from the corresponding result of the RAD re-runs since the temporal evolution of the surface temperature differs."*

has been amended as follows:

*"This result is based on the original coupled runs, and differs from the corresponding result of the RAD re-runs since the temporal evolution of the surface temperature differs although the finding that polar amplification asymmetry decreases markedly in the flat AA model setup holds in the RAD re-runs as well"*

Furthermore, Fig. R1 of this reply was added as Fig. 5 to the revised manuscript in order to better explain a result that is not robust across the sets of runs and minor clarifications have been added to the description of the result (on page 8 lines 7 to 10 of the revised manuscript). Fig. R1 should be compared to Fig. 5 of the original (Fig. 4 of the revised) manuscript.

The point that the OHT changes are not robust has also been re-iterated in Sect. 3.6 on page 11 lines 17ff of the revised manuscript as follows:

*"Finally, in Sect. 3.3 it was shown that small OHT changes in response to doubling $CO_2$ across the Arctic Circle that contribute to the OHT difference in Fig. 12 depend on the whether the original runs or the RAD re-runs are analyzed."*

[Figure]

Figure R1: Oceanic heat transport (OHT) as in Fig. 5 of the original manuscript but for the original run.

Please note that the non-robustness of this result is reflected by a difference between Fig. R1 and Fig. 5 of the original manuscript that appears to be very small.

Although a set of two runs (i.e. the original and the RAD re-runs) is a very small "ensemble", and although the RAD re-runs are short since they are much slower to run due to increased I/O and more computationally expensive than the original runs, the general similarities between the results of the RAD-reuruns and the original coupled runs indicate that the main conclusions are robust between these two sets of runs. At the same time a comparison between the two sets of runs gives a first indication as to where the robustness is limited.

In order to investigate whether the stronger (or faster) antarctic warming in the flat AA run is mainly related to ocean transients, the base control and 2xCO2 runs and the corresponding two flat AA runs from 200 to 600 years

[Figure]

Figure R2: (a) 25-year running mean of the temperature difference between the 2xCO2 and the control runs for the arctic and the antarctic region in the base and the flat AA model setup and (b) ratio of arctic to antarctic amplification based on 25 year running mean time series.

and Sect. 3.7 has been expanded to discuss these runs. Fig R2a (which corresponds to Fig. 14a of the revised manuscript) shows that antarctic warming in the flat AA run is stronger than arctic warming almost throughout the entire 600 year period. The figure also shows that the antarctic temperature increase eventually becomes larger than the arctic temperature increase in both model setups. This point (which is in part explained by a MOC slowdown in both

model setups and in part by a steady antarctic warming in both model setups) is discussed in Sect. 3.7 of the revised manuscript (page 12, lines 23ff) and also in Section 2 (page 3, lines 28ff).

Furthermore, the revised manuscript discusses potential improvements of the current model setup in Sect. 3.7 which are re-iterated in lines 16ff of the the abstract and at the end of the conclusion section. One of the improvements that is suggested is to use a proper ensemble (if possible of several high resolution models).

2) Discussion of the role of sfc vs. atmospheric temperatures The author convincingly shows that the change in sfc temperature and its relationship to atmospheric warming is causing changes in the lapse rate feedback, rather than the warming profile within the atmosphere. However, it is not clear to me that the temperature feedback should therefore be regarded as a single mechanism. This point might benefit from either rethinking or more detailed discussion.

I partially agree with this criticism and I have adapted a corresponding statement in the conclusion section and added a discussion to the end of Section 3.5 (see details below).

Here, the main reason for lumping the two feedbacks is simply that the change of the LR feedback was not associated with a difference in atmospheric temperature. Furthermore, as you have noted above, the lumped feedback has a simple physical interpretation as it corresponds to the the total thermal feedback.

In order to point out that lumping the feedbacks is not uncommon, I have added the sentence

*"The definition of the TA feedback is identical to the definition of the long-wave feedback in a study by Winton et al. (2006) and to the definition of the temperature feedback in a study by Block and Mauritsen (2013)."*

to page 10, lines 20ff of the revised manuscript.

On the other hand, decomposing the total thermal feedback into LR and PL feedbacks is indeed very useful for understanding polar climate change, and I appreciate that your comment gives me the opportunity to clarify this point.

I have replaced the following sentence of the conclusions:

*"On the other hand, unlike for the tropics in the polar regions there appears to be no clear rationale for separating the total temperature feedback into the Planck and the lapse rate feedback."*

by:

*"Although the rationale for decomposing the total temperature feedback into the Planck and the lapse rate feedback is less clear in the polar regions than in the tropics, such a decomposition is nevertheless useful for understanding polar climate change (see e.g. Pithan and Mauritsen, 2014; Payne et al., 2015). A more detailed discussion of this issue was given at the end of Sect. 3.5."*

The newly added discussion at the end of Section 3.5 reads as follows:

*"The usual decomposition of the total temperature feedback into PL and LR feedbacks is nevetheless useful for understanding polar climate change. As explained above, the PL feedback is defined as the hypothetical feedback that would be expected if the atmosphere would warm at the same rate as the surface. However, the polar atmosphere generally warms less than the surface due to a lack of vertical mixing (e.g. Manabe and Wetherald, 1975). The tropical atmosphere, on the other hand, warms more than the surface. Therefore, in order to radiate away a given amount of energy a larger surface warming is required in the polar regions compared to the tropics (Pithan and Mauritsen, 2014). The lack of atmospheric warming in the polar atmosphere relative to*

*the surface is reflected in the large positive polar lapse rate feedback."*

Furthermore, I have changed the following sentence on page 9, lines 12ff of the original manuscript (page 10, lines 16ff of the revised manuscript) that describes the rationale for lumping the LR and the PL feedback.

*"The rationale for this is that the sum LR+PL is mainly sensitive to changes in surface temperature and not to changes in the atmospheric lapse rate above the surface layer."*

to:

*"The rationale for this is that in the present study setup the sum LR+PL in the antarctic region is mainly sensitive to changes in surface temperature and not to changes in the atmospheric lapse rate above the surface layer."*

Minor issues:

l. 5 if→when (If + past tense triggers would in main clause, and indicates a hypothetical case) This issue reappears in the manuscript.

Changed "if" to "when" in on page 1, line 5, page 6, line 29, and on page 7, line 20 of the revised manuscript.

l.6 (and elsewhere): please reserve "significant" for its statistical meaning, and specify the statistical test.

"Significantly" was replaced by "notably" on page 1, line 6, page 2, line 2, and page 13, line 7 of the revised manuscript. On page 1 in line 11 "significant" was replaced by "considerable".

p.2 l1 ff: What is the baseline for warming?

Added *"relative to a 1986-2005 reference period"*.

[Figure]

Figure R3: Single plot

Corrected. Thank you very much.

p. 2 l.32, general issue: I would suggest to consistently speak of either polar amplification symmetry or asymmetry to avoid confusion.

Changed "increasing polar amplification symmetry" to "decreasing polar amplification asymmetry' throughout the revised manuscript. I also corrected "decrease" to "increase" on p. 15, l. 17.

Added *"by the antarctic surface height"*.

Corrected. Thank you very much.

Figure 1 and 2: I suggest combining these into one Figure, or at least using the same axes to facilitate comparisons.

Figure R3 of this reply shows Figures 1 and 2 of the original manuscript combined into a single plot and Figure R4 shows a panel containing Figures 1 and

[Figure]

Figure R4: Panel plot

2 with identical y-axis scaling. Figure R3 shows that in the northern hemisphere more solar radiation is absorbed and terrestrial radiation emitted in the low-resolution CESM v1.0.6 run compared to CERES SYN1deg Edition 3A. A part (but certainly not all) of this difference is probably due to comparing a pre-industrial run to near present day observations (which are influenced by higher aerosol concentrations in the northern hemisphere and also land use changes). Nevertheless, based on the magnitude of the difference, other model biases almost certainly dominate. In particular, one could speculate that the model bias might be related to the representation of clouds over land and sea ice. Regardless of what the reason for this bias might be, a discussion of this bias seems not overly relevant to the arctic/antarctic asymmetry discussed in the manuscript. The bias does, however, make Figure R3 look rather busy. Consequently, the simulated decrease in asymmetry between the two polar regions when Antarctica is assumed to be flat is easier to see in Figure R4 than

in Figure R3. Therefore, unless you strongly prefer Figure R3, I would rather replace Figures 1 and 2 by Figure R4.

[revised manuscript text omitted]

**3.3 Atmosphere and ocean heat transport**

In this section, the zonal mean AHT (Fig. 3) and OHT (Fig. 4) for the years 80-109 are compared between the base and the flat AA model setup and their changes in the corresponding $CO_2$ doubling runs are analyzed based on the RAD re-runs. As expected, based on Fig. 1a, AHT and OHT were more symmetric in the flat AA model setup than in the base setup (although they can not be expected to become completely symmetric because the overall land mass distribution differs between the northern and the southern hemisphere). In the control runs, the poleward AHT increased in the flat AA model setup compared to the base setup mainly in the southern hemisphere. Poleward OHT [A.Rev. #1 and T.C.]across the polar circles increased in the southern hemisphere and also slightly increased in the northern hemisphere. In the original coupled runs, on the other hand, poleward OHT [A.Rev. #1 and T.C.]across the polar circle in the northern hemisphere [A.Rev. #1 and T.C.]slightly decreased during this period [A.Rev. #1 and T.C.](Fig. 5)(not shown). 
[revised manuscript text omitted]
$_2$ A.Rev. #1 and T.C.(from the RAD re-runs).**Please note: modified y-axis label in lower panel.**

[Figure]

**Figure 4.** Same as Fig. 3 but for oceanic heat transport (OHT).**Please note: modified y-axis label in lower panel.**

[Figure]

**Figure 5.** A.Rev. #1 and T.C.Same as Fig. 4 but from the original runs instead of the RAD re-runs.**Please note: newly added figure.**

[Figure]

**Figure 6.** $CO_2$ radiative forcing (CO2) as well as Planck (PL), lapse rate (LR), water vapor (WV), albedo (ALB), and cloud (CL) radiative feedback based on two-sided PRP calculations for years 80–109. FSU=LR+WV+ALB+CL is the sum of the feedbacks except PL. The residual (RES) is the difference between the radiative perturbation from replacing all variables simultaneously and the sum (SUM=CO2+LR+WV+ALB+CL+PL) of the individual contributions including CO2 and PL. **Please note: modified y-axis labels.**

[Figure]

**Figure 7.** Differences between Fig. 6 (a) and (b). **Please note: modified y-axis label.**

[Figure]

**Figure 8.** Radiative forcing and feedback maps for the base model setup for the years 80–109. Labels as in Fig. 6.

[Figure]

**Figure 9.** Differences between the flat AA and base the setup (comparable to Fig. 8).

[Figure]

**Figure 10.** Average air temperature profiles for the antarctic and the arctic region from the coupled runs and differences (2xCO2 minus Control) for the base and the flat AA model setup for years 80–109. Dots and crosses denote average surface air temperature at the region average surface pressure. Dots correspond to solid and crosses to dashed lines. Only pressure levels where more than 20% of the grid points are above the ground are shown in the profiles. **Please note: modified x-axis label in (b) and (d).**

[Figure]

**Figure 11.** ^(Anonym. Rev. #1) Lapse rate (LR) and Planck (PL) feedback in the base and the flat AA model setup and for the sensitivity calculations described in Table 2. ^(T. Cronin) All computations are based on PRP calculations using 73-hourly instantaneous model output and the original base and flat AA runs are analyzed. **Please note: modified y-axis label**.

[Figure]

**Figure 12.** Difference flat AA minus base setup of ^(T. Cronin) $(F_{AR} - F_{AA})$ , where ^(T. Cronin) $(F_{AR} - F_{AA})$  is the difference between the arctic and the antarctic region shown in Fig. 7. For the forcing and the feedbacks, this corresponds to the difference between the red and the blue bars in Fig. 7. FSUP=PL+LR+WV+ALB+CL is the sum of all feedbacks including the Planck feedback. It is compared to contributions from atmospheric and oceanic heat transport (AHT, OHT) and heat storage (HS). The residual (RES) is defined as the difference between HS and the sum of forcing, feedbacks and heat transport convergence (CO2+FSUP+AHT+OHT). **Please note: modified y-axis label.**

[Figure]

**Figure 13.** Difference between the zonal mean surface air temperature and the time averaged zonal mean surface air temperature from the corresponding control run ($T_{av}$) in the control and the 2xCO2 run for the base and the flat AA model setup. **Please note: extended runs by 400 years and modified color bar label.**

[Figure]

**Figure 14.** [A.Rev. #1 and T.C.] (a) 25-year running mean of the temperature difference between the 2xCO2 and the control runs for the arctic and the antarctic region in the base and the flat AA model setup and (b) ratio of arctic to antarctic amplification based on 25 year running mean time series. **Please note: newly added figure**

**Table 1.** Coupled Runs

| run | description |
|---|---|
| base Control | Control run for constant year 1850 conditions |
| base 2xCO2 | Same as base Control, but doubled $CO_2$ concentration |
| flat Antarctica Control | Control run for flat Antarctica (flat AA) |
| flat Antarctica 2xCO2 | flat AA $CO_2$ doubling run |

**Table 2.** Additional PRP Sensitivity Computations

| label | variable(s) from flat AA model setup in base model setup |
|---|---|
| LRPLSens | surface air ($T_s$) and atmospheric ($T_a$) temperature |
| LRsens | atmospheric temperature $T_a$ |
| PLSens | $T_s$ and control $T_a$ with added $\Delta T_s$ as in PL |

**Table 3.** Mean arctic and antarctic warming (2xCO2 minus Control) $\pm$ one standard deviation for three consecutive 25-year time [T. Cronin] slices starting at year 51 for the base and the flat AA model setup.

| | Arctic Warming (K) | Antarctic Warming (K) | Difference |
|---|---|---|---|
| base | $3.95 \pm 0.48$ | $2.55 \pm 0.22$ | $1.40 \pm 0.29$ |
| flat AA | $3.76 \pm 0.31$ | $3.17 \pm 0.01$ | $0.59 \pm 0.32$ |

---

## Author Comment (AC2) · 14 Mar 2017

**Response to Prof. Timothy Cronin**

I very much thank Prof. Cronin for insightful and constructive comments and for a very thorough review of the manuscript. Prof. Cronin's comments are repeated in dark red and the replies are typeset in black. Appended to this response is a draft version of a revised manuscript including track changes (the editor will decide later whether a revised manuscript should be submitted). For brevity, this will be referred to simply as "revised manuscript".

Major Comments:

1) One major issue is that the ocean circulation may change considerably between flat-AA and base setups, and it may not be in pre-industrial equilibrium in either case. A hint of this is seen in Figure 5, and a trace of its implications may also emerge in figure 10. How long was the model run for to get a stable "preindustrial climate" in both the flat-AA and base setups? If the ocean is not equilibrated and I doubt it would be given that most runs appear to occur for ~100-200 years and this is an order of magnitude less time than required for equilibration how do we definitively interpret differences between flat-AA and base setups as being results of changes in terrain and not ocean transients? Some issues with this are hinted at in section 3.6, where southern hemispheric heat content changes are noted as large, and also in section 3.7, where the MOC starts to collapse in the 2xCO2 run and leads to changes in Arctic temperature that may be purely due to internal variability rather than forcing or changes in topography.

The control base run was started from a spun-up state. As indicated in Fig. 13c of the original manuscript, the surface temperature in the noAA control run changed little after the initial decades (as expected based on several studies that showed that a large part of the surface temperature response to an initial perturbation occurs in the first decades while the gradual adjustment of the

deep ocean takes much longer). Furthermore, there is a flat AA control run and a flat AA 2xCO2 run in which the deep ocean was still adjusting gradually to the change in surface height, so that taking the difference is expected to remove most of this effect.

In order to investigate whether the stronger antarctic warming in the flat AA runs is due to ocean transients, the original coupled sensitivity runs have been extended from 200 to 600 years. The results are shown in Figs. R1 and R2 of this response which correspond to Figs. 13 and 14 of the revised manuscript.

The extended coupled sensitivity runs show that the antarctic warming is stronger in the flat AA setup compared to the standard setup almost throughout the entire 600 year period (Fig. R2a). Based on this result, I find it very unlikely that the stronger antarctic warming in the flat AA run is due to ocean transients. Note also, that abrupt $CO_2$ doubling experiments have often been used in climate research even though the models were very seldomly run to equilibrium.

Regarding the MOC slowdown it was noted in the original manuscript that a slowdown of the MOC due to $CO_2$ doubling was also found in other climate models. The results from the extended runs in Figs. R1 and R2 further illustrate the point that such a slowdown is fairly common.

In the extended base 2xCO2 model run, the MOC starts to slow down around year 200 and then starts to gradually recover during the last 200 years of the 600 year run (Fig. R1b). In the flat AA model run the slowdown occurs earlier (moderate weakening starts around year 100 and the strongest slowdown occurs after year 120, see Fig. R1d). After 200 years antarctic warming is larger than arctic warming not only in the flat AA but also in the base model setup. While this does not contradict present-day observations or CMIP5 model results, it clearly differs from present-day observations of a stronger polar amplification in the arctic compared to the antarctic region and also from results of

[Figure]

Figure R1: As Fig 13 of the original manuscript, but for extended runs.

shorter CMIP5 future scenario runs. As explained in the original manuscript, the findings regarding MOC slowdown should not be overinterpreted since the MOC tends to react more sensitively to $CO_2$ increases in low-resolution models compared to high-resolution models, and additional caveats with respect to using this setup for projecting future antarctic climate change (especially the lack of an ice sheet model) have been highlighted in Sect. 3.7 of the revised manuscript.

[Figure]

Figure R2: (a) 25-year running mean of the temperature difference between the 2xCO2 and the control runs for the arctic and the antarctic region in the base and the flat AA model setup and (b) ratio of arctic to antarctic amplification based on 25 year running mean time series.

The choice of time slices has been better motivated in the methods section of the revised manuscript as follows (see also my response to point C below):

*The analysis focuses on the transient response to $CO_2$ doubling during the years 76 to 125 and especially during the years 80 to 109 for which the model was re-run and PRP calculations were performed. While most of the temper-*

*ature response in the upper ocean to the $CO_2$ perturbation takes place during the initial decades of a 2xCO2 perturbation experiment the deep ocean has not yet reached equilibrium during this period which is similar to the situation in present-day climate and transient future climate change. In the base 2xCO2 run antarctic warming (see Sect. 3.7) became stronger than arctic warming around year 200 (and around year 120 for the flat AA 2xCO2 run) which differs from present-day observations of a stronger polar amplification in the arctic compared to the antarctic region and from the results of shorter CMIP5 simulations which the present study aims to help explain.*

Sect. 3.7 has been partially re-written and extended to describe the extended runs and to include the main arguments from above. Potential improvements to the model setup have also been proposed in Sect. 3.7. It now reads as follows (please see the appended manuscript for complete track changes):

"*Fig. 13 shows the temporal evolution of the zonal mean surface temperature in the control runs and the 2xCO2 runs for the original coupled runs. As expected, the largest surface temperature response to assuming Antarctica to be flat occurred during the first decades of the flat AA control run (Fig. 13c). These decades were not taken onto account in the preceding analysis. After this, the surface temperatures remained fairly stable in the flat AA control run, although the deep ocean was still adjusting. Taking the difference between the flat AA 2xCO2 run and the flat AA control run (as was done in the preceding sections) is expected to remove most of the latter effect.*

*Fig. 13b and d show that antarctic surface temperatures increased faster in the flat AA 2xCO2 run than in the base 2xCO2 run as expected based on the previous sections. The arctic temperatures varied strongly due to internal variability, which helps to explain the differences between the original coupled run and the coupled re-runs with half-hourly radiation calls which have been pointed out in the discussion of the ocean heat transport in Sect. 3.3.*

*The weaker arctic warming in the middle of the 2xCO2 base run (Fig. 13b) is an indication of a slowing of the ocean's meridional overturning circulation (MOC). Such a slowdown has often been found in $CO_2$ perturbation experiments, and it tended to be stronger in low-resolution low-complexity models compared to state-of-the art high-resolution models. Since the CESM was run at a low resolution in this study, this finding should also not be overinterpreted. In the 2xCO2 flat AA run, the MOC started to slow down earlier than in the 2xCO2 base run (Fig. 13d), which might indicate that assuming a flat Antarctica did not only influence the Antarctic region but also the arctic region. For a more reliable estimate of this effect coupled runs at a higher resolution and an ensemble of model runs with slightly perturbed initial conditions would be required.*

*Fig. 14a shows the evolution of the arctic and antarctic surface temperature for the base and the flat AA model setup and Fig. 14b shows ratios of the arctic to the antarctic amplification which are computed as:*

$$f = \frac{A_{Ar}}{A_{AA}} = \frac{\hat{T}_{Ar,2xCO2} - \hat{T}_{Ar,Control}}{\hat{T}_{AA,2xCO2} - \hat{T}_{AA,Control}}$$

*where $\hat{T}$ is a regional average 25-year running mean temperature.*

*It should be noted that antarctic warming relative to the respective control run (Fig. 14a) was stronger in the flat AA than in the base model setup almost throughout the entire 600 year period. However, even though the temperature in the flat AA control run stabilized after a moderate initial warming and even though the temperature evolution from the control run was subtracted in this analysis, it can not be completely ruled out that this moderate initial warming could have also played a role in the later development in the 2xCO2 flat AA run. Therefore, in retrospect, starting the flat AA 2xCO2 from a separate long flat AA spinup run and prescribing a more realistic gradual increase of the $CO_2$ concentration which would allow to also inspect the first decades of the*

*CO2 perturbation experiments would have been better.*

*After 250 years, f was lower than unity in the base and the flat AA model setup which indicates that the antarctic temperature increase was stronger than the arctic temperature increase in both model setups. This finding is related to a slowdown of the north Atlantic MOC in both of the runs which could in part be a transient feature as the MOC recovery times are known to be extremely long. Again, in order to gain confidence in this result, additional ensemble model runs at higher resolution and ideally also a comparison with results from a multi-model ensemble would be necessary. Since the aim of the various model sensitivity runs has been to investigate the sensitivity of the polar amplification asymmetry to antarctic surface height (and not to provide a future projection of antarctic climate change under global warming), the runs were performed without an ice sheet model. In order to arrive at a more credible projection of antarctic climate change, state-of-the art high resolution models that include state-of-the art ice sheet dynamics models should be used.*

The suggestions for improving the model setup have been re-iterated at the end of the conclusion section of the revised manuscript as follows:

*"Potential future studies based on coupled climate model runs that aim to study the influence of surface elevation on the polar amplification asymmetry which is found in present-day observations and also in future climate projections (which often span only one or one and a half centuries) would benefit from a separate long flat AA spinup run and from prescribing a more realistic gradual increase of the $CO_2$ concentration."*

And the following sentence was added to the end of the abstract of the revised manuscript:

*"In order to arrive at a more reliable estimate of the role of land height for the observed polar amplification asymmetry, additional studies based on*

*ensemble runs from higher resolution models and an improved model setup with a more realistic gradual increase of the $CO_2$ concentration are required."*

Based on the newly extended flat AA control run, performing an additional CO2 doubling run would take two or three additional weeks (depending on availability of computing resources). The PRP runs, on the other hand take several months to run.

2) I find the multiple layers of differences to make some figures, discussion, and notation very confusing. There are essentially three sets of differences used in this paper: a climate change difference (2xCO2 − 1xCO2), a hemispheric difference (Arctic – Antarctic), and a model setup difference (Flat-AA - base). Each of these is at some point annotated as $\Delta$: in Figures 3-5 and 10, a single $\Delta$ means a climate change signal, in Figures 7 and 11, a single $\Delta$ refers to a Arctic-Antarctic difference, and in figure 9 a single $\Delta$ refers to Flat-AA minus base difference. In the worst case, all three are used together in figure 12 as $\Delta(\Delta F)$, or the difference (flat-AA – base) between Arctic and Antarctic feedbacks (with feedbacks F being in turn a difference between 1x and 2x CO2 radiative fluxes). I would suggest using subscripts for all but one of these differences, e.g., T2x-T1x for climate change in response to doubling CO2, and FA-FAA for hemispheric difference, and reserving $\Delta$ for Flat-AA minus base, as it is least-easily subscripted (though $F_- - F\hat{}$ could serve here too).

I have replaced $\Delta F$ in the y-axis label of Fig. 6 of the revised manuscript (which is also Fig. 6 of the original manuscript) by $F_{Ar}$-$F_{AA}$ and adapted the labels in Figs. 6, 11, and 12 of the revised manuscript accordingly. For the differences 2xCO2 minus 1xCO2, I have added the index "2x" to the $\Delta$ in the y-axis labels of the corresponding plots except in the new figure 14 (for an example see Fig. R3). In this case, I found that spelling out the differences does not increase readability very much (see Figs. R4 and R5). I have also

[Figure]

Figure R3: Modified y-axis label in Fig. 2 of the revised manuscript (Fig. 3 of the original manuscript).

replaced the $\Delta T$ in the color bar label of Fig. 12 of the revised manuscript by $(T - T_{av})$ and explained that "av" denotes a time average in the caption. $\Delta$ (without index) is now reserved for flat AA minus base as suggested.

The caption of Fig. 11 of the revised manuscript was also modified to better explain the plot. The original caption

*Lapse rate (LR) and Planck (PL) feedback in the base and the flat AA model setup and for the sensitivity calculations described in Table 2.*

was changed to (see also comment regarding 73-hourly output below):

*Arctic minus Antarctic difference of the lapse rate (LR) and Planck (PL) feedback in the base and the flat AA model setup and for the sensitivity calculations described in Table 2 (based on PRP calculations using 73-hourly instantaneous model output).*

3) The definition of "feedbacks" as changes in TOA radiative fluxes, unnormalized by surface temperature changes, complicates interpretation of findings in this paper for me. I am not sure what the basis is for this choice is, and I think it could be better explained if there is some strong reason for us-

[Figure]

Figure R4: Using differences in the y-axis label in Fig. 2 of the revised manuscript.

[Figure]

Figure R5: Using differences in the y-axis label in Fig. 2 of the revised manuscript.

ing these changes rather than the standard of feedbacks being normalized by temperature change  e.g., $\lambda = (R2x\ R1x)/(T2x\ T1x)$ (units of W m-2 K-1). In Figures 6 and 7, especially, I find that use of changes in TOA fluxes rather than standard feedbacks convolutes changes in polar warming with changes in the true per-unit warming feedbacks $\lambda$. For example, do the Planck and lapse-rate "feedbacks" in Figure 6 change just because there is more warming in the Antarctic in the flat-AA case, or because each $\lambda$ has also changed? Use of the conventional feedback metric normalized by surface temperature change would also allow for more straightforward comparison to previous work.

Radiative feedbacks in general circulation models (GCMs) are most commonly defined as TOA flux changes which are normalized by the global mean surface temperature change and not by the local temperature change. Because the entire forcing/feedback formalism is usually based on an equation for the global mean energy budget, normalization by the local temperature change would induce major conceptual problems in a GCM context. In this case, horizontal advection terms would have to be included in the budget equation. These terms are not negligible especially in the polar regions and they are not constant under climate change (and they might also depend on the nature of the forcing). Therefore, in GCM studies TOA radiation flux changes are usually normalized by global mean temperature even when plotting maps. Fig. 1 shows that unlike the tropics, the polar regions are nowhere near to either radiative or radiative-convective equilibrium and that instead a radiative-advective equilibrium assumption as in Payne et al., GRL, 2015, Cronin and Jansen, GRL, 2016, clearly is more appropriate.

Here, the feedbacks are defined per $CO_2$ doubling with respect to 1850. Thus, one can directly compare the magnitudes of the forcing and the feedbacks. Furthermore, both magnitudes can then also be compared to transport and heat storage terms. As stated in the sentence starting in the last line on page

4 of the original manuscript, this choice also facilitates a straight forward comparison of the actual TOA energy flux differences associated with the base and the flat AA model setup. Since the sole purpose of the PRP analysis is to try to find out what causes differences in warming between the arctic and antarctic region in the different model setups, to me it seems reasonable to compare the actual TOA flux differences without normalizing by the global mean temperature change. Furthermore, even though instantaneous 2xCO2 perturbations are used, the manuscript focuses on the transient response. As noted in on page 5 in lines 2f of the original manuscript, a normalization by surface temperature change is not appropriate in this case.

4) The reasons for the RAD re-runs are not made completely clear, and it is not clear where they are used and where they are not used. This is a particular issue around line 30 on page 6, where you say "the original coupled runs were chosen for this analysis" but then in the next paragraph, you say that the simulations behind figure 3 will be analyzed in more detail  but the caption of figure 3 says that they are RAD re-runs.

The following sentences have been added to page 4, lines 14ff of the revised manuscript in order to more explicitly explain why two sets of runs are used:

*"The advantage of the RAD re-runs is that the PRP analysis is more exact which makes them more suitable for a budget analysis. The advantage of the base model setup is that it is computationally cheaper to run and thus is better suited for longer integrations."*

The fact that the rad re-runs instead of the original coupled runs are used in several of the following sections was addressed by adding the following statement on page 7, lines 31f of the revised manuscript:

*"... based on the RAD-reruns which yield sufficiently similar results to the original runs for this analysis to be useful."*

[Figure]

Figure R6: As Fig. 6 of the original (and the revised) manuscript, but based on the original coupled runs with 73-hourly instantaneous model output.

The captions of Figs. 3 and 11 have been augmented by stating which runs were used.

Fig. R6 is based on the original runs with 73-hourly instantaneous output. Originally, I had hoped that performing the PRP offline radiation computations at this low frequency for a fairly long period would yield sufficiently exact results (which would have meant much quicker runs, much smaller amounts of data, and consequently also a much smaller logistic challenge). Only later I found out that not only the output frequency was to low but that also the frequency of the radiation calls had to be increased in order to increase the accuracy of the PRP computations.

The differences between Fig. R6 and Fig. 6 of the original (and also the revised) manuscript are due to a combination of internal variability (as the RAD-reruns trigger new realizations) and the more exact budgets (for which the radiation has been called every time step instead of every other time step and the PRP computations have been performed for three-hourly instead of 73-hourly instantaneous model output). The decision to perform the RAD-reruns was triggered by an analysis of the reasons behind the large "RES" term in Fig. R6.

Although two sets of runs (i.e. the original and the RAD re-runs) is not a very impressive ensemble, the general similarities between the results indicate that the main conclusions are robust and at the same time to some extend also suggests where the robustness is limited. In order to better explain one of the results that differs between the RAD-reruns and the original coupled runs, Fig. R7 of this reply was added as Fig. 5 to the revised manuscript (see also my response to major comment #1 by reviewer #1.

5) [Self-promotion disclosure!] Some work that I have been involved with recently has, I think, helped provide a better understanding of the high-latitude lapse-rate feedback. You might want to look at Payne, Jansen, and Cronin (2015), GRL, "Conceptual model analysis of the influence of temperature feedbacks on polar amplification" and Cronin & Jansen (2016), GRL, "Analytic radiative-advective equilibrium as a model for high-latitude climate". Both papers attempt to develop a better prognostic understanding of the high-latitude lapse-rate feedback, which, as you note in section 3.5, is mostly connected to near-surface temperature changes being larger than mid- and upper-tropospheric temperature changes. Figures 3 and 4 of Payne et al (2015) are similar to your Figure 10, except that we focused on low-latitude/high-latitude differences and not polar asymmetries, and we were using simpler column models to get a sense of temperature structure changes. I have also

[Figure]

Figure R7: Oceanic heat transport (OHT) as in Fig. 5 of the original manuscript but for the original coupled runs.

done some preliminary work on polar amplification asymmetry due to surface height with a radiative-advective equilibrium model (nowhere near publication, so certainly nothing to cite!). If you plan to work on the subject more, it might be good to talk outside the review process, and potentially collaborate.

The two published studies mentioned above are clearly relevant for the general discussion regarding polar feedbacks. In the revised manuscript both are cited in the introduction. The Payne et al., 2015 paper is also in the "conclusions" section of the revised manuscript (see details below).

I am currently not planning to work much more on this particular topic (i.e. the polar amplification asymmetry), but I have done some very limited initial work on another question regarding polar warming. I would certainly appreciate a discussion either on the polar amplification asymmetry or on some other topic related to polar warming outside the review process in case you are interested.

Other specific comments:

A) On page 5, lines 8 and 10  what are the respective polar circles, exactly? Its not obvious that you can choose 66.55 N and S when the models resolution is only T31  so exact limits of integration and averaging should be noted more clearly.

I added the following sentence on page 5, lines 31f of the revised manuscript:

*"The corresponding grid cell edges of the atmosphere grid are located at 66.8° North and South. Oceanic heat transport is diagnosed at 66.6° North and South."*

B) I suggest combining Figures 1 and 2; it is difficult to compare the simulations to observations at present and an additional set of black lines in 1) should be completely readable.

A similar suggestion was made by the other reviewer (although the other reviewer also indicated that using identical axis on the two plots might be sufficient). In order to avoid having to switch back and forth between documents, I repeat my response here.

Figure R8 of this reply shows Figures 1 and 2 of the original manuscript combined into a single plot and Figure R9 shows a panel containing Figures 1 and 2 with identical y-axis scaling. Figure R8 shows that in the northern hemisphere more solar radiation is absorbed and terrestrial radiation emitted in the low-resolution CESM v1.0.6 run compared to CERES SYN1deg Edition 3A. A part (but certainly not all) of this difference is probably due to comparing a pre-industrial run to near present day observations (which are influenced by higher aerosol concentrations in the northern hemisphere and also land use changes). Nevertheless, based on the magnitude of the difference, other model

[Figure]

Figure R8: Single plot

[Figure]

Figure R9: Panel plot

biases almost certainly dominate. In particular, one could speculate that the model bias might be related to the representation of clouds over land and sea ice. Regardless of what the reason for this bias might be, a discussion of this bias seems not overly relevant to the arctic/antarctic asymmetry discussed in

the manuscript. The bias does, however, make Figure R8 look rather busy. Consequently, the simulated decrease in asymmetry between the two polar regions when Antarctica is assumed to be flat is easier to see in Figure R9 than in Figure R8. Therefore, unless you strongly prefer Figure R8, I would rather replace Figures 1 and 2 by Figure R9.

C) Regarding the three time slices and standard deviations in Table 3 and p6, lines 17–27. I dont understand why these three time slices were chosen, or what the % of warming difference explained by topography means, exactly. And how are the +/- standard deviations determined  based on three points (the means of the three time slices), or based on each year treated as an independent point?

In order to better motivate the choice of the analysis time the following sentences were added to the methods section of the revised manuscript (compare also my response to major comment 1 above):

*The analysis focuses on the transient response to $CO_2$ doubling during the years 76 to 125 and especially during the years 80 to 109 for which the model was re-run and PRP calculations were performed. While most of the temperature response in the upper ocean to the $CO_2$ perturbation takes place during the initial decades of a 2xCO2 perturbation experiment the deep ocean has not yet reached equilibrium during this period which is similar to the situation in present-day climate and transient future climate change. In the base 2xCO2 run antarctic warming (see Sect. 3.7) became stronger than arctic warming around year 200 (and around year 120 for the flat AA 2xCO2 run) which differs from present-day observations of a stronger polar amplification in the arctic compared to the antarctic region and from the results of shorter CMIP5 simulations which the present study aims to help explain.*

In order to clarify that the mean and standard deviation refer to the mean and standard deviation of the three time slices and also in order to better explain how the fraction of the warming difference that is explained by surface height

is computed, the following sentences:

*In the base run, the arctic region warmed on average by 1.40 ± 0.29 K (mean ± one standard deviation) more than the antarctic region. In the flat AA run, on the other hand, the difference between the Arctic and the antarctic warming was reduced to 0.59 ± 0.32 K. On average, about 56 ± 30% of the difference in warming was explained for the three time slices.*

were changed to:

*In the base run, the arctic region warmed on average by 1.40 ± 0.29 K (mean of three time slices ± one standard deviation) more than the antarctic region. In the flat AA run, on the other hand, the difference between the arctic and the antarctic warming was reduced to 0.59 ± 0.32 K. Thus, on average, about 56 ± 30% of the difference in warming between the arctic and the antarctic region (i.e. 0.81 ± 0.46 K of 1.40 ± 0.29 K) was explained by the antarctic surface height for the three time slices.*

The sentence:

*If only antarctic temperature change were taken into account, 73%, 26%, and 42% were explained by antarctic surface height.*

was replaced by:

*When only antarctic temperature change was taken into account (i.e. the arctic temperature increase was taken from the base setup while the antarctic temperature increase was taken from the flat AA setup), 73%, 26%, and 42% were explained by antarctic surface height.*

A correction replacing "time" by "time slices" was added to the caption of table 3.

D) In Figures 6 and 7: whether or not you stick to calling changes in radiative fluxes feedbacks or you decide to normalize them by temperature changes and

make them true feedbacks, I think it would help to include changes in AHT and OHT convergences here, as well as storage if it is significant. Or, you could mention that the SUM term in each plot adds up to the sum of changes in AHT convergence, OHT convergence, and storage. It surprises me that the changes in heat flux convergence would be so small relative to local feedbacks. I do appreciate your note on p8 L5 that the values should not be over-interpreted as heat transports vary rapidly near the margins of the Arctic and Antarctic caps

I have added sentences regarding the magnitude of the AHT and OHT convergence and heat storage relative to the SUM term to the revised manuscript (see below).

One reason for not including the changes in AHT and OHT convergence and the heat storage term in figure 6 of the original manuscript (which is also figure 6 of the revised manuscript) was that they are individually of the same order of magnitude as the sum of the feedbacks (SUM) and that the bars would thus be very short. Furthermore, as mentioned on page 5 in lines 16 to 22 of the original manuscript and then re-iterated in on page 9 in lines 20 to 24 of the original manuscript, the changes in AHT convergence, OHT convergence, and heat storage which are diagnosed as described in Section 2 do not add up to the sum of the SUM and the RES term in Fig. 6 as one might expect. Since this "budget imbalance" (which might be either due to inaccuracies in the analysis method or in the model's energy balance as discussed in Section 2) does not depend strongly on the model setup, it cancels out in the analysis presented in Section 3.6 (as indicated by the small RES term in Fig. 12 of the original and the revised manuscript). This is why I decided to include the analysis in Section 3.6 in spite of the problems with computing a closed budget that are mentioned in Section 3.6 and in Section 2.

The following sentences were added to page 8, lines 26ff of the revised manuscript:

*"The SUM and the RES terms in Fig. 6 are expected to balance contributions*

*from changes in AHT convergence, OHT convergence, and heat storage. The corresponding individual contributions were diagnosed separately as explained in Section 2 and they were found to be of the same order of magnitude as the SUM term which is much smaller than the individual contributions of the major feedbacks except CL (not shown). Unfortunately, however, as explained in Section 2, the diagnosed contributions do not add up to the SUM+RES term as expected."*

Because the polar regions are approximately in advective-radiative equilibrium (Payne et al., GRL, 2015, Cronin and Jansen, GRL, 2016, see also Fig. 1), it is indeed interesting to see the large degree of cancellation between the contributions from the individual feedbacks suggested by Fig. 6 of the revised manuscript. On the other hand, it is also well understood that a decreasing meridional temperature gradient that is associated with polar amplification acts to decrease meridional heat transport convergence.

Even though the SUM term is small compared to the individual feedbacks, the subsequent analysis in Section 3.6 suggests that contributions from ocean and atmosphere heat transport convergence and heat storage are important for explaining the polar amplification asymmetry.

In addition to the changes outlined above, where appropriate I have also changed "*transport*" to "*change in transport convergence*" since the reviewer used the term "transport convergence" in his comments and since I also find it useful to make this distinction. Transport is often used to describe either a (net) flux (which with respect to an arbitrary region could in principle also mean "transit") or a flux divergence and it had previously been used for both.

E) Figure 10: The mismatch between the profiles and dots/crosses is confusing here. Why is there so much less change in the crosses in b) and d) than in the profiles at lowest-levels? Why is the red dot in c) warmer than the profile at

lowest levels? And why are the base and flat-AA profiles so different in c)?

The following sentences explaining the reason for the mismatch were added to page 9, lines 28ff of the revised manuscript:

*"The apparent mismatch between the dots and crosses and the profiles at the lowest atmospheric levels in Fig. 9 (b) and (d) is explained by the condition that all pressure levels where more than 20% of the grid points are above ground are shown in the profiles. Consequently, only a limited number of grid points entered the average temperature for the lowest atmospheric levels while the average surface temperature was computed as an average over all grid points."*

F) Page 8, paragraph around line 25: I suggest eliminating Table 2  I found it more confusing than helpful  and relocating the paragraph from the methods section describing these simulations to this point in section 3.5 (paragraph from p4 L20-25). The information is not necessary when presented at the time in section 2, and will be more helpful in this section if the reader does not need to flip back to the earlier description.

I disagree on this point. I find that the methods section is right the place to consult when looking up information on the methods. I also think Table 2 is useful.

G) Page 8 Line 34-Page 9 line 2, re: "weaker LR feedback in the flat AA model setup" I thought Figure 6 showed the LR feedback was stronger in the flat-AA model setup? Do you mean that the LR feedback asymmetry is smaller in the flat-AA model setup (shown in Figure 7)?

Indeed. It must read *stronger* and not *weaker*. Thank you very much.

H) Page 9, lines 3-9. I'm a little puzzled by these assertions and the choice

to lump the Planck and LR feedbacks together. I agree that your sensitivity tests have shown that the lapse rate feedback is dominated by changes in surface temperature rather than atmospheric temperature, or that the "lapse-rate feedback" is a feedback related to how surface temperatures and near-surface temperatures change relative to the mid- and upper-troposphere (essentially, the lapse rate feedback relates to changes in the strength of surface inversions). Thus, it still seems meaningful to me to separate the lapse rate and Planck feedbacks, since the Planck feedback could be calculated given the models control state alone, whereas changes in inversion strength are a complicated result of changes in many linked model processes. [Also, note: I do not see any references to Figure 11 in this section (or elsewhere). Perhaps the reference disappeared at the end of p9 L5? Figure 11 did not seem like the most compelling support for your argument; I think looking at each feedback by itself rather than their hemispheric asymmetry would make a stronger point]

A similar concern was raised by Reviewer #1. In order to avoid having to switch forth back between documents, I repeat my arguments here.

The main reason for lumping the two feedbacks is simply that the change of the LR feedback was not associated with a difference in atmospheric temperature. Furthermore, the lumped feedback has a simple physical interpretation as it corresponds to the the total thermal feedback (see page 9 lines 14f of the original manuscript).

In order to point out that lumping the feedbacks is not uncommon, I have added the sentence

*"The definition of the TA feedback is identical to the definition of the longwave feedback in a study by Winton et al. (2006) and to the definition of the temperature feedback in a study by Block and Mauritsen (2013)."*

to page 10, lines 20ff of the revised manuscript.

On the other hand, decomposing the total thermal feedback into LR and PL feedbacks is indeed very useful for understanding polar climate change, and I appreciate that your comment gives me the opportunity to clarify this point.

I have replaced the following sentence of the conclusions:

*"On the other hand, unlike for the tropics in the polar regions there appears to be no clear rationale for separating the total temperature feedback into the Planck and the lapse rate feedback."*

by:

*"Although the rationale for decomposing the total temperature feedback into the Planck and the lapse rate feedback is less clear in the polar regions than in the tropics, such a decomposition is nevertheless useful for understanding polar climate change (see e.g. Pithan and Mauritsen, 2014; Payne et al., 2015). A more detailed discussion of this issue was given at the end of Sect. 3.5."*

The newly added discussion at the end of Section 3.5 reads as follows:

*"The usual decomposition of the total temperature feedback into PL and LR feedbacks is nevertheless useful for understanding polar climate change. As explained above, the PL feedback is defined as the hypothetical feedback that would be expected if the atmosphere would warm at the same rate as the surface. However, the polar atmosphere generally warms less than the surface due to a lack of vertical mixing (e.g. Manabe and Wetherald, 1975). The tropical atmosphere, on the other hand, warms more than the surface. Therefore, in order to radiate away a given amount of energy a larger surface warming is required in the polar regions compared to the tropics (Pithan and Mauritsen, 2014). The lack of atmospheric warming in the polar atmosphere relative to the surface is reflected in the large positive polar lapse rate feedback."*

Furthermore, I have changed the following sentence on page 9, lines 12ff of the original manuscript (page 10, lines 16ff of the revised manuscript) that describes the rationale for lumping the LR and the PL feedback.

*"The rationale for this is that the sum LR+PL is mainly sensitive to changes in surface temperature and not to changes in the atmospheric lapse rate above the surface layer."*

to:

*"The rationale for this is that in the present study setup the sum LR+PL in the antarctic region is mainly sensitive to changes in surface temperature and not to changes in the atmospheric lapse rate above the surface layer."*

Based on the first paragraph of the discussion and conclusion section in Cronin and Jansen (2016), I have the impression that separating the thermal feedback into LR and PL feedback might be related to what you call the "nonuniquess of the lapse rate feedback". But I am not sure whether I am interpreting this right?

Regarding the reference to Fig. 11: thank you for spotting this missing reference. I have added a reference to this figure on page 10 in line 2 of the revised manuscript.

I) Relatedly, on p 11, lines 1-4: I dont really follow this argument, as noted above. I also disagree with this assertion that there is no reason to separate the high-latitude Planck and lapse rate feedbacks. The reason for low-latitude separation is that we can calculate them separately, basically from first principles, if stratification follows a moist adiabat and we know the control-state climate and further, that there is cancellation between water vapor and LR feedbacks in the tropics. That we don't understand the high-latitude lapse rate feedback fully seems to be a very good reason not to lump it in with a feedback that we do understand better (the Planck feedback).

I have addressed this issue in my response to point H above. Please refer to this response.

p1 L 6; p6 L13; p7 L10  should be "led to" if in the past tense

Changed from "lead to" to "led to" following the access review. I hope that this is what you meant.

p1, and then recurring: I am unsure of the correct style guidelines for capitalization of "Arctic", "Antarctic", and phrases containing these two terms. I have (perhaps) erred on the side of always capitalizing. Super minor issue, will presumably be fixed at the proof level, but I am interested in the "right" answer!

Merriam-Webster appears to be fine with either spelling for the adjective. It seems that except in some place names like "Arctic Ocean" they default to the lower case version (e.g. "arctic sea smoke", "arctic air"). The online version of the Cambridge Dictionary suggests that the adjective is "Arctic" in British English and "arctic" in American English. The Associated Press web site says that they have a new entry for "arctic char" in the food section of their style guide, but they might actually be more concerned about the correct spelling of the "char(r)" than the "arctic". Several American newspapers capitalize the adjective and based on a quick Google search it could even be that capitalization is more common in "Arctic amplification" than in "arctic air mass" (no real statistics here). I certainly don't have an opinion on this and I am fine with whatever the copy editors or the reviewers suggest.

p 1 L19  may want to put numbers on "substantially", e.g., "2-3 times as much as the global average"

The *substantially* refers to the absolute warming. It is quantified on page 2 in line 1 of the original manuscript (page 2, line 5 of the revised manuscript).

**p2 L4** suggest rewording to "The focus in explaining arctic . . . has long been on the . . ." (rather than starting sentence with "For long")

Done.

**p2 L9; p8 L17** You could mention work from point 5) above at either of these locations.

I have mentioned both studies in the introduction on page 2, line 14 of the revised manuscript. Furthermore, I have mentioned the Payne et al., 2015 study in the "Conclusions" section on page 13 in lines 23f of the revised manuscript.

**p2 L26** should be "water vapor feedback" (not just "vapor feedback")

Changed.

**p2 L27** should be "Antarctic" and "land-sea distribution"

Changed.

**p2 L28** suggest changing "play a role for. . ." to "play a role in . . ."

Done. I also changed "role for" to "role in" on page 3 in line 5 and on page 13 in line 4 of the revised manuscript.

**p3 L19-21** suggest changing "every second time step" to "every other time step" (as seconds are units of time

Done.

**p4 L16** suggest hyphenating three-hourly

Done (following access review).

p4 L21-22 what does "73 hourly instantaneous model output" mean? (This looks like a typo to me)

A lower output frequency was used. I have replaced "73 hourly" by 73-hourly" on page 5 line 8 of the revised manuscript and also added the following sentence to page 5, line 12ff of the revised manuscript:

*"The lower output frequency in the additional sensitivity computations was used since it significantly reduces storage requirements and run time although the PRP computations become less accurate."*

The lower accuracy is evidenced by the larger residual (RES) in Figure R6 of this reply and also by the larger PL difference in Fig. 11 of the original and the revised manuscript compared to Fig. 6.

Please note, that unlike the standard model runs, the PRP method is very I/O intensive. Compared to running the model in its standard configuration with standard I/O, running the model with increased I/O frequency and then performing offline PRP computations using takes many times as long. The PRP runs presented in this manuscript were performed on two dedicated multi-core dual processor servers with standard external SAS storage raids attached via a file server over a period of several months.

Done.

the reader here by saying something about the size of the imbalance.

I have looked around the available documentation and also inspected parts of the code, but I did not find an indication of an added heat source in the Arctic. One feature that could in principle cause a local energy imbalance in the model are the energy and mass fixers that are applied in the atmosphere model which are meant to ensure global energy and mass balance. As a side effect, these mass and energy fixers induce something like a non-local transport that can cause local budgets to be out off balance. Then again, it is not clear that the present problem is indeed related to a local energy imbalance inside the model. If it is, it would certainly require a major model development effort to to pin down the root cause(s) and to devise alternative formulations to ensure local conservation. A number of potential candidates for explaining such an energy imbalance exist and as far as I know it would be much more difficult to ensure local conservation in a spectral model compared to a grid point model. In fact, the prospect of more readily achieving local energy balance (which requires that the transport of water vapor and hydrometeor mass is locally mass conserving) is one reason behind the move toward grid point models in climate research, although at low resolution spectral models still maintain advantages. The very existence of global mass and energy fixers in the low resolution atmosphere model used in this study provide a hint of the difficulties that are associated with ensuring local mass and energy conservation (because local conservation implies global conservation).

Regarding the magnitude of the budget deficit, the following sentence has been added to page 6 lines 14ff of the revised manuscript:

*"The imbalance between net radiation deficit in the polar regions and the sum of the other budget terms (dominated by poleward heat transports) varies between 2 and 5% of the net radiation deficit."*

When analyzing the differences between the 2xCO2 and the control runs, the imbalance (included in the SUM term in Fig. 6) is of comparable magnitude to the change of the transport terms (also included in the SUM term in Fig. 6; compare my response to specific point D above). Both are much smaller than the main local feedbacks which are also shown in Fig. 6. When looking at these differences, the contribution from the imbalance for the Arctic is larger than that for Antarctica (which cannot be seen in Fig. 6). However, because the imbalance appears to be systematic (i.e. the contribution from the imbalance is independent of whether the base or the flat AA model setup is used), the budget analysis is nevertheless considered useful. In particular, the residual that is shown in Fig. 12 of the revised manuscript (which is also Fig 12 of the original manuscript) is small compared to the other terms.

p5 L31; p6 L4 suggest deleting "rather" in these sentences, as similar is already qualitative

Deleted rather on page 6 in lines 25 and 30 of the revised manuscript where it was used together with similar and also on page 9 in line 25 of the revised manuscript.

p7 L7-9 re: "Poleward OHT increased in the southern hemisphere and also slightly increased in the northern hemisphere" and subsequent sentence Figure 5 seems to me to show a decrease in poleward OHT in the NH. . . are you showing the "original coupled run" results in Figure 5? (this ties to comment 4 above)

Thank you for pointing this out. The sentence has been revised as follows:

*"Poleward OHT across the polar circles increased in the southern hemisphere and also slightly increased in the northern hemisphere."*

In order to illustrate the difference between the original runs and the RAD

reruns, Fig. 5 was added to the revised manuscript.

**p7 L13** No comma after "Both"

Removed comma after "Both" on page 8 line 14 of the revised manuscript.

**p7 L29-30** The interpretation of reduced polar feedback asymmetry is complicated by the feedbacks being expressed as radiative flux changes rather than true feedbacks per unit temperature change (see point 3 above)

Please refer to my reply to point #3 above.

**p8 L3** suggest changing to "was not directly affected by. . ."

Changed *"not affected"* to *"not directly affected"* on page 9 in line 9 of the revised manuscript.

**p9 L 30** Paragraph around here makes an excellent point about the semi-arbitrary extent of polar caps.

Thank you very much.

**p10 L8, 12** suggest hyphenating "half-hourly" "low-complexity", and "high-resolution"

Done. For consistency, I also hyphenated "low-resolution" throughout the revised manuscript.

[revised manuscript text omitted]

30    Sect. 3.7, the reasons for the [Anonym. Rev. #1]decreased polar amplification [Anonym. Rev. #1]asymmetry in the flat AA model setup (Fig. 2) will be analyzed in some detail [A.Rev. #1 and T.C.]based on the RAD re-runs which generally yield sufficiently similar results to the original runs for this analysis to be useful.

**3.3 Atmosphere and ocean heat transport**

In this section, the zonal mean AHT (Fig. 3) and OHT (Fig. 4) for the years 80-109 are compared between the base and the flat AA model setup and their changes in the corresponding $CO_2$ doubling runs are analyzed based on the RAD re-runs. As expected, based on Fig. 1a, AHT and OHT were more symmetric in the flat AA model setup than in the base setup (although they can not be expected to become completely symmetric because the overall land mass distribution differs between the northern and the southern hemisphere). In the control runs, the poleward AHT increased in the flat AA model setup compared to the base setup mainly in the southern hemisphere. Poleward OHT [A.Rev. #1 and T.C.]across the polar circles increased in the southern hemisphere and also slightly increased in the northern hemisphere. In the original coupled runs, on the other hand, poleward OHT [A.Rev. #1 and T.C.]across the polar circle in the northern hemisphere [A.Rev. #1 and T.C.]slightly decreased during this period [A.Rev. #1 and T.C.](Fig. 5)(not shown). 
[revised manuscript text omitted]
$_2$ A.Rev. #1 and T.C.(from the RAD re-runs).**Please note: modified y-axis label in lower panel.**

[Figure]

**Figure 4.** Same as Fig. 3 but for oceanic heat transport (OHT).**Please note: modified y-axis label in lower panel.**

[Figure]

**Figure 5.** A.Rev. #1 and T.C. Same as Fig. 4 but from the original runs instead of the RAD re-runs.**Please note: newly added figure.**

[Figure]

**Figure 6.** CO₂ radiative forcing (CO2) as well as Planck (PL), lapse rate (LR), water vapor (WV), albedo (ALB), and cloud (CL) radiative feedback based on two-sided PRP calculations for years 80–109. FSU=LR+WV+ALB+CL is the sum of the feedbacks except PL. The residual (RES) is the difference between the radiative perturbation from replacing all variables simultaneously and the sum (SUM=CO2+LR+WV+ALB+CL+PL) of the individual contributions including CO2 and PL. **Please note: modified y-axis labels.**

[Figure]

**Figure 7.** Differences between Fig. 6 (a) and (b). **Please note: modified y-axis label.**

[Figure]

**Figure 8.** Radiative forcing and feedback maps for the base model setup for the years 80–109. Labels as in Fig. 6.

[Figure]

**Figure 9.** Differences between the flat AA and base the setup (comparable to Fig. 8).

[Figure]

**Figure 10.** Average air temperature profiles for the antarctic and the arctic region from the coupled runs and differences (2xCO2 minus Control) for the base and the flat AA model setup for years 80–109. Dots and crosses denote average surface air temperature at the region average surface pressure. Dots correspond to solid and crosses to dashed lines. Only pressure levels where more than 20% of the grid points are above the ground are shown in the profiles. **Please note: modified x-axis label in (b) and (d).**

[Figure]

**Figure 11.** Anonym. Rev. #1 Arctic minus Antarctic difference of the lLapse rate (LR) and Planck (PL) feedback in the base and the flat AA model setup and for the sensitivity calculations described in Table 2. T. CroninAll computations are based on PRP calculations using 73-hourly instantaneous model output and the original base and flat AA runs are analyzed. **Please note: modified y-axis label**.

[Figure]

**Figure 12.** Difference flat AA minus base setup of T. Cronin$(F_{AR} - F_{AA})\Delta F$, where T. Cronin$(F_{AR} - F_{AA})\Delta F$ is the difference between the arctic and the antarctic region shown in Fig. 7. For the forcing and the feedbacks, this corresponds to the difference between the red and the blue bars in Fig. 7. FSUP=PL+LR+WV+ALB+CL is the sum of all feedbacks including the Planck feedback. It is compared to contributions from atmospheric and oceanic heat transport (AHT, OHT) and heat storage (HS). The residual (RES) is defined as the difference between HS and the sum of forcing, feedbacks and heat transport convergence (CO2+FSUP+AHT+OHT). **Please note: modified y-axis label.**

[Figure]

**Figure 13.** Difference between the zonal mean surface air temperature and the time averaged zonal mean surface air temperature from the corresponding control run ($T_{av}$) in the control and the 2xCO2 run for the base and the flat AA model setup. **Please note: extended runs by 400 years and modified color bar label.**

[Figure]

**Figure 14.** [A.Rev. #1 and T.C.] (a) 25-year running mean of the temperature difference between the 2xCO2 and the control runs for the arctic and the antarctic region in the base and the flat AA model setup and (b) ratio of arctic to antarctic amplification based on 25 year running mean time series. **Please note: newly added figure**

**Table 1.** Coupled Runs

| run | description |
|---|---|
| base Control | Control run for constant year 1850 conditions |
| base 2xCO2 | Same as base Control, but doubled $CO_2$ concentration |
| flat Antarctica Control | Control run for flat Antarctica (flat AA) |
| flat Antarctica 2xCO2 | flat AA $CO_2$ doubling run |

**Table 2.** Additional PRP Sensitivity Computations

| label | variable(s) from flat AA model setup in base model setup |
|---|---|
| LRPLSens | surface air ($T_s$) and atmospheric ($T_a$) temperature |
| LRsens | atmospheric temperature $T_a$ |
| PLSens | $T_s$ and control $T_a$ with added $\Delta T_s$ as in PL |

**Table 3.** Mean arctic and antarctic warming (2xCO2 minus Control) $\pm$ one standard deviation for three consecutive 25-year time [T. Cronin] slices starting at year 51 for the base and the flat AA model setup.

| | Arctic Warming (K) | Antarctic Warming (K) | Difference |
|---|---|---|---|
| base | $3.95 \pm 0.48$ | $2.55 \pm 0.22$ | $1.40 \pm 0.29$ |
| flat AA | $3.76 \pm 0.31$ | $3.17 \pm 0.01$ | $0.59 \pm 0.32$ |